# Virological characteristics of the SARS-CoV-2 XBB variant derived from recombination of two Omicron subvariants

Tomokazu Tamura [1,2,41], Jumpei Ito [3,41], Keiya Uriu[3,4,41], Jiri Zahradnik [5,6,41], Izumi Kida [7,41], Yuki Anraku [8,41], Hesham Nasser [9,10,41], Maya Shofa[11,12,41], Yoshitaka Oda[13,41], Spyros Lytras [14,41], Naganori Nao[15,16], Yukari Itakura [2,17], Sayaka Deguchi[18], Rigel Suzuki[1,2], Lei Wang [13,19], MST Monira Begum[9], Shunsuke Kita [8], Hisano Yajima[20], Jiei Sasaki[20], Kaori Sasaki-Tabata[21], Ryo Shimizu[9], Masumi Tsuda [13,19], Yusuke Kosugi[3,4], Shigeru Fujita [3,4], Lin Pan[3,22], Daniel Sauter [3,23], Kumiko Yoshimatsu [24], Saori Suzuki [1,2], Hiroyuki Asakura[25], Mami Nagashima[25], Kenji Sadamasu[25], Kazuhisa Yoshimura[25], Yuki Yamamoto[26], Tetsuharu Nagamoto[26], Gideon Schreiber [5], Katsumi Maenaka [2,8,27,28], The Genotype to Phenotype Japan (G2P-Japan) Consortium*, Takao Hashiguchi [20,29], Terumasa Ikeda [9], Takasuke Fukuhara [1,2,30,31], Akatsuki Saito [11,12,32], Shinya Tanaka [13,19] ✉, Keita Matsuno [2,7,16,33] ✉, Kazuo Takayama[18,29] ✉ & Kei Sato [3,4,22,29,34,35,36] ✉

In late 2022, SARS-CoV-2 Omicron subvariants have become highly diversified, and XBB is spreading rapidly around the world. Our phylogenetic analyses suggested that XBB emerged through the recombination of two cocirculating BA.2 lineages, BJ.1 and BM.1.1.1 (a progeny of BA.2.75), during the summer of 2022. XBB.1 is the variant most profoundly resistant to BA.2/5 breakthrough infection sera to date and is more fusogenic than BA.2.75. The recombination breakpoint is located in the receptor-binding domain of spike, and each region of the recombinant spike confers immune evasion and increases fusogenicity. We further provide the structural basis for the interaction between XBB.1 spike and human ACE2. Finally, the intrinsic pathogenicity of XBB.1 in male hamsters is comparable to or even lower than that of BA.2.75. Our multiscale investigation provides evidence suggesting that XBB is the first observed SARS-CoV-2 variant to increase its fitness through recombination rather than substitutions.

The SARS-CoV-2 Omicron variant has been the current variant of concern since the end of 2021[1]. As of December 2022, recently emerging Omicron subvariants are undergoing convergent evolution, acquiring substitutions at the same residues of the spike (S) protein, such as R346, K444, L452, N460, and F486[2,3]. For instance, the Omicron BQ.1.1 variant, which is a descendant of Omicron BA.5 and is becoming predominant in Western countries[1] as of December 2022, possesses all convergent substitutions, such as R346T, K444T, L452R, N460K, and F486V. Recent studies, including ours, have suggested that L452R[4–9], N460K[2,6,10,11], and R346T[2] increase the binding affinity of the SARS-CoV-2 S protein to human angiotensin-converting enzyme 2 (ACE2), the receptor for viral infection, while R346T[12,13], K444T[13] and F486V[2,4,5,13–15] contribute to evasion of antiviral humoral immunity induced by vaccination and natural SARS-CoV-2 infection. Similar to the observations

A full list of affiliations appears at the end of the paper. *A list of authors and their affiliations appears at the end of the paper.
✉e-mail: tanaka@med.hokudai.ac.jp; matsuk@czc.hokudai.ac.jp; kazuo.takayama@cira.kyoto-u.ac.jp; KeiSato@g.ecc.u-tokyo.ac.jp

in BA.5[5] and BA.2.75[10], combinational substitutions in the S protein to (1) evade antiviral humoral immunity in exchange for a decrease in ACE2 binding affinity (e.g., F486V) and (2) enhance ACE2 binding affinity to compensate for the decreased affinity associated with immune evasion substitution (e.g., L452R and N460K) have been frequently observed in recently emerging Omicron subvariants, including BQ.1.1. These observations suggest that acquiring these two types of substitutions in the S protein is a trend that allows recently emerging Omicron subvariants to spread more efficiently than prior ones.

In addition to the diversification and subsequent convergent evolution of emerging Omicron subvariants (e.g., BQ.1.1), a recombinant variant called XBB has recently emerged. The Omicron XBB variant likely originated through the recombination of two BA.2 descendants, BJ.1 and BM.1.1.1 a progeny of BA.2.75[16]. While the BQ.1 lineage is becoming predominant in Europe, XBB has become predominant in India and Singapore and is spreading in several countries[17] as of December 2022. As of October 28, 2022, the WHO classifies XBB as an Omicron subvariant under monitoring[1]. Recent studies including ours have revealed the virological features of BQ.1[2,12,18]. However, the features of XBB, another Omicron subvariant of concern, have not been fully elucidated.

In this study, we explored the virological characteristics of XBB, particularly its transmissibility, immune resistance, ACE2 binding affinity, infectivity, fusogenicity, structural information and pathogenicity in a hamster model without a history of vaccination and viral infection (hereafter referred to as intrinsic pathogenicity).

## Results

### Evolution and epidemics of the XBB variant

As of December 2022, most of the prevalent Omicron lineages, including BA.5, belong to the phylogenetic clade related to BA.2 (Fig. 1a). Of these, certain highly diversified BA.2 subvariants, such as BA.2.75 and BJ.1, were first identified in South Asia and are referred to as second-generation BA.2 variants (Fig. 1a). Recently, the XBB variant emerged as a recombinant lineage between the second-generation BA.2 variants BJ.1 (BA.2.10.1.1) and BM.1.1.1 (BA.2.75.3.1.1.1; a descendant of BA.2.75)[16] (Fig. 1a). XBB harbors the substitutions R346T, N460K, and F486S, which were convergently acquired during Omicron evolution (Fig. 1b; the mutations in the non-S region are summarized in Supplementary Fig. 1)[2]. To trace the recombination event that led to the emergence of the XBB variant, we retrieved all SARS-CoV-2 sequences deposited to GISAID (as of October 3, 2022) with PANGO lineage designation matching BJ.1, BM.1, XBB, and all their descendant lineages (including BM.1.1, BM.1.1.1, and XBB.1). Recombination analysis on the aligned set of sequences, using a number of independent recombination detection methods implemented in RDP5[19] (see "Methods"), robustly identified a single recombination breakpoint unique to all XBB sequences at genomic position 22,920 (matching the Wuhan-Hu-1 reference genome) (Fig. 1c). No evidence of recombination was found in the BJ.1 and BM.1 sequences in the dataset. Consistent with the result of the RDP5 analysis, visual inspection of the nucleotide differences between the consensus sequences of XBB, BJ.1, and BM.1 (including BM.1.1 and BM.1.1.1) clearly illustrated that the identity of XBB to BJ.1 ends at genome position 22,942, and the identity of XBB to BM.1 starts after position 22,896 (Fig. 1c). Together, our analysis suggests that the recombination breakpoint is between positions 22,897 and 22,941, within the receptor binding domain (RBD) of the S protein (corresponding to amino acid positions 445–460) (Fig. 1c).

We then split the sequence alignment at position 22,920 to determine the evolutionary history of each nonrecombinant segment of the XBB genomes. The phylogenetic reconstructions recapitulate the recombination results, with the 5′ end major parental sequence being derived from the BJ.1 clade and the 3′ end minor parental sequence from the BM.1.1.1 clade (Fig. 1d). Using the longer 5′ end nonrecombinant part of these genomes, we estimated the emergence

date of XBB using Bayesian tip-dated phylogenetic inference (see "Methods") (Fig. 1d). Our analysis suggests that the XBB clade's most recent common ancestor (tMRCA) existed at the start of July 2022 (median posterior date: July 7, 2022; 95% HPD confidence intervals: from June 10, 2022, to July 29, 2022). We also dated the tMRCA between the XBB and BJ.1 lineages at the start of June 2022 (median posterior date: June 11, 2022; 95% HPD intervals: from May 22, 2022, to June 26, 2022) (Fig. 1d). Together, our analyses suggest that XBB emerged through the recombination of two cocirculating lineages, BJ.1 and BM.1.1.1, during the summer of 2022.

To trace the shift in viral fitness during the evolution of Omicron that led to the emergence of XBB, we estimated the effective reproduction number ($R_e$) of XBB-related variants based on the epidemic data of SARS-CoV-2 in India, where XBB-related lineages circulated (from June 1 to November 15, 2022) (Fig. 1e and Supplementary Fig. 1b and Supplementary Table 1). BJ.1 and BM.1/BM.1.1/BA.1.1.1 showed higher $R_e$ values than their parental lineages, BA.2.10 and BA.2.75, respectively. Furthermore, the $R_e$ value of XBB is 1.23- and 1.20-times higher than those of the parental BJ.1 and BM.1.1.1, respectively (Fig. 1e and Supplementary Fig. 1b and Supplementary Table 1). Importantly, this is the first documented example of a SARS-CoV-2 variant increasing its fitness through recombination rather than substitutions.

As of December 2022, two viral lineages are expanding their epidemics around the world: BQ.1 lineages and XBB lineages. To investigate the prevalence of these two lineages in various geographic regions, we estimated the epidemic frequency of each variant as of November 15, 2022, in each county (Fig. 1f and Supplementary Table 2). BQ.1 lineages have spread and reached dominance in European, American, and African countries, reflecting the likelihood that BQ.1 emerged from the African continent[20] (Fig. 1f). On the other hand, XBB lineages have spread and reached dominance in South and Southeast Asian countries, such as India, Singapore, and Indonesia, reflecting the fact that XBB was first identified in South Asia (Fig. 1f). Furthermore, we constructed a hierarchical Bayesian model and estimated the global average and country-specific $R_e$ values of XBB lineages according to the epidemic data of countries where XBB lineages cocirculated with BQ.1 lineages (Fig. 1g, h, Supplementary Fig. 1c and Supplementary Table 3). Our analysis shows that the $R_e$ values of XBB and XBB.1 (i.e., XBB harboring S:G252V) are 1.24- and 1.26-times higher than that of BA.5 and are comparable with those of BQ.1 and BQ.1.1 (Fig. 1g and Supplementary Fig. 1c). Together, our analyses show that both BQ.1 and XBB lineages, which exhibit similar advantages in estimated viral fitness, are becoming predominant in the Western and Eastern regions of the world, respectively.

### Immune resistance of XBB.1

To investigate the virological features of XBB, we first evaluated the immune resistance of XBB using HIV-1-based pseudoviruses. In the present study, we used the major S haplotype of XBB lineages as of October 3, 2022, corresponding to the S protein of XBB.1, for the following experiments. In the case of breakthrough BA.2 infection sera, BA.2.75 did not exhibit significant resistance when compared to BA.2 (Fig. 2a), which is consistent with our prior study[10]. In contrast, we found that XBB.1 exhibits profound (30-fold) resistance to breakthrough BA.2 infection sera ($P = 0.0002$, Fig. 2a). To determine the amino acid substitutions conferring this resistance to breakthrough antisera, we constructed BA.2 S mutants that harbor individual single substitutions present in XBB.1. We did not analyse the substitutions that also appear in BA.2.75 (e.g., G446S) since we already analysed these substitutions in our previous study[10]. As shown in Fig. 2a, several substitutions, such as V83A (2.1-fold, $P = 0.0034$), Y144del (2.9-fold, $P = 0.0002$), Q183E (2.0-fold, $P = 0.0039$), R346T (2.1-fold, $P = 0.0005$), L368I (1.8-fold, $P = 0.042$), V445P (2.1-fold, $P = 0.0002$), F486S (3.0-fold, $P = 0.0002$), and F490S (2.7-fold, $P = 0.024$), conferred significant resistance to breakthrough BA.2 infection sera. Because the immune

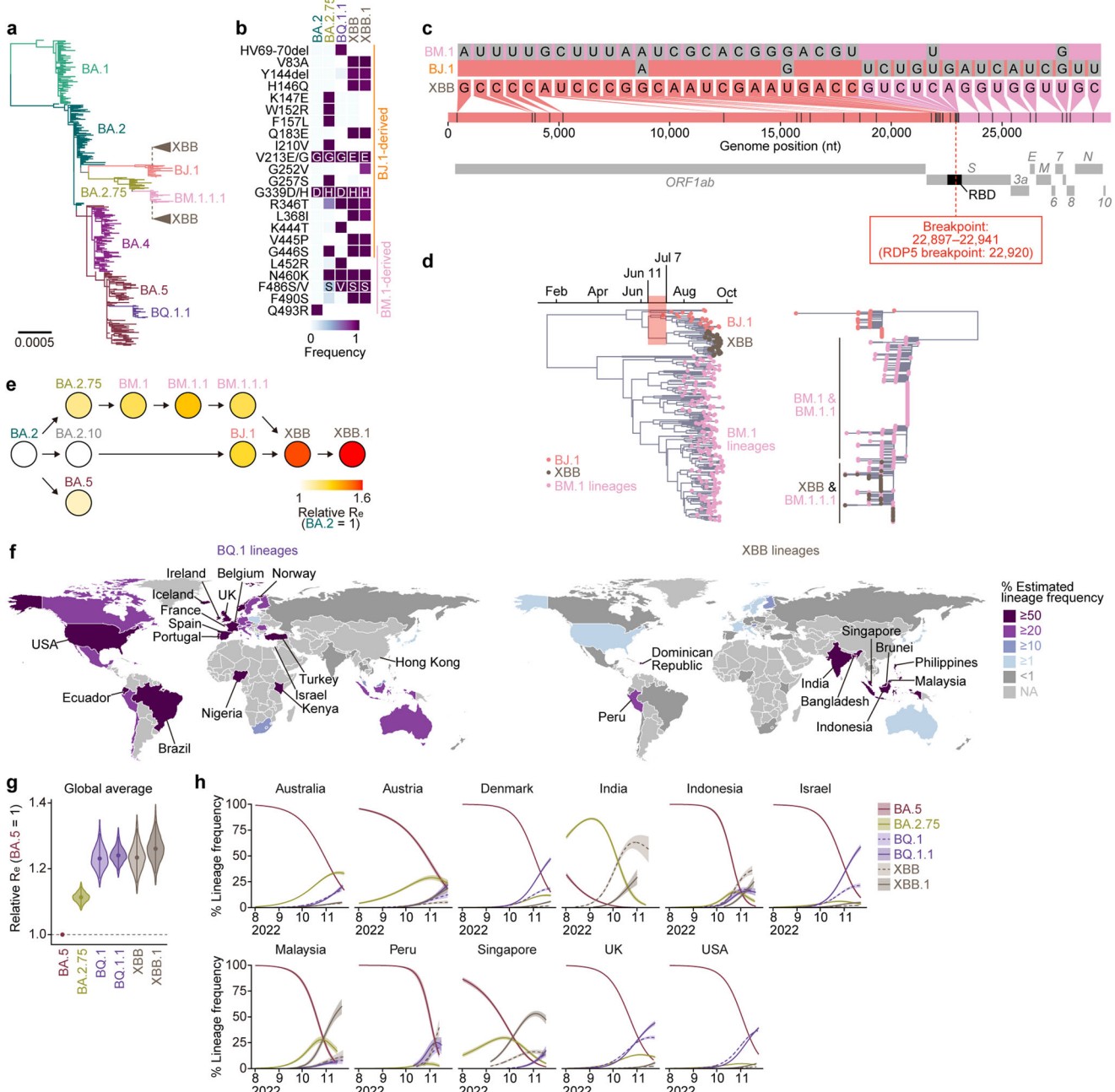

**Fig. 1 | Phylogenetic and epidemic analyses of the XBB lineage. a** Maximum likelihood tree of representative sequences from PANGO lineages of interest: BA.1, BA.2, BA.4, BA.5, BA.2.75, BJ.1 and BM.1.1.1, rooted on a B.1.1 outgroup (not shown). The recombinant parents of XBB are annotated on the tree as cartoon clades. **b** Amino acid differences in the S proteins of Omicron lineages. **c** Nucleotide differences between the consensus sequences of the BJ.1, BM.1 (including BM.1.1/ BM.1.1.1) lineages and the XBB (including XBB.1) lineage, visualized with snipit (https://github.com/aineniamh/snipit). **d** Maximum clade credibility time-calibrated phylogeny of the 5′ non-recombinant segment (1–22,920) of the XBB variant (left) and non-calibrated maximum likelihood phylogeny of the 3′ non-recombinant segment (22,920–29,903) (right). The right hand-side tree is rooted on a BA.2 outgroup (not shown). **e** Relative effective reproduction number ($R_e$)

values for viral lineages in India, assuming a fixed generation time of 2.1 days. The $R_e$ of BA.2 is set at 1. Dot color indicates the posterior mean of the $R_e$, and an arrow indicate phylogenetic relationship. See also Supplementary Fig. 1b. **f** Difference in the circulated regions between BQ.1 and XBB lineages. Estimated lineage frequency as of November 15th, 2022 in each country is shown. Countries with ≥50% and ≥20% frequencies are annotated for the BQ.1 and XBB lineages, respectively. **g** Relative $R_e$ values for viral lineages, assuming a fixed generation time of 2.1 days. The $R_e$ value of BA.5 is set at 1. The posterior (violin), posterior mean (dot), and 95% Bayesian confidential interval (CI; line) are shown. The global average values estimated by a hierarchical Bayesian model[27] are shown. See also Supplementary Fig. 1c. **h** Estimated lineage dynamics in each country where BQ.1 and XBB lineages cocirculated. Posterior mean, line; 95% CI, ribbon. Source data are provided with this paper.

resistance conferred by each individual substitution is relatively minor when compared to the resistance of XBB.1 (Fig. 2a), our data suggest that multiple substitutions in the XBB.1 S cooperatively contribute to the resistance against humoral immunity induced by breakthrough BA.2 infection.

Consistent with our previous study[10], BA.2.75 showed a statistically significant (1.8-fold) resistance to breakthrough BA.5 infection sera when compared to BA.2 (*P* = 0.0016, Fig. 2b). Moreover, XBB.1 exhibited profound (13-fold) resistance to breakthrough BA.5 infection sera (*P* < 0.0001, Fig. 2b). A neutralization assay using pseudoviruses

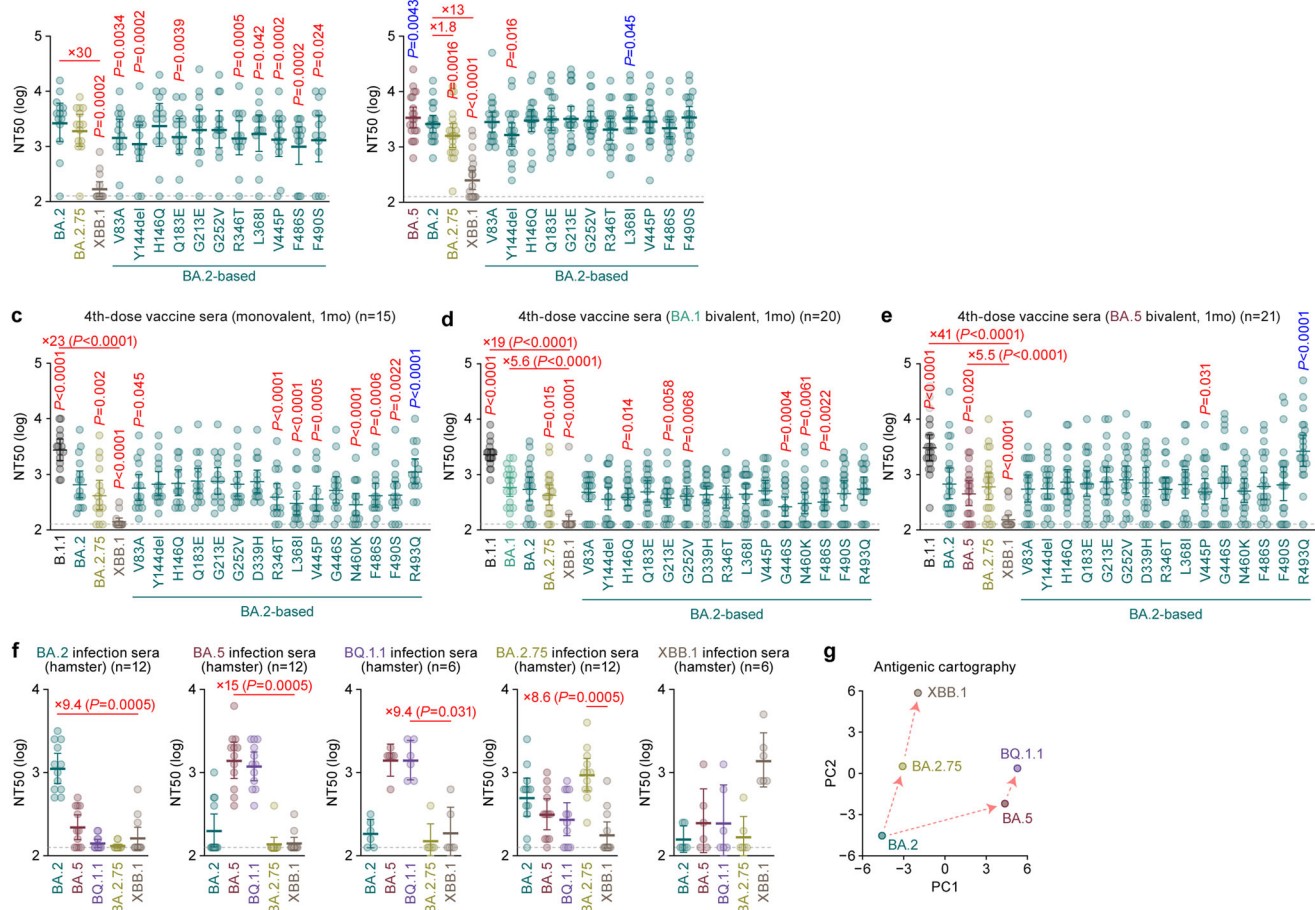

**Fig. 2 | Immune resistance of XBB.1.** Neutralization assays were performed with pseudoviruses harboring the spike (S) proteins of B.1.1, BA.1, BA.2, BA.5, BQ.1.1, BA.2.75 and XBB.1. The BA.2 S-based derivatives are included in (**a–e**). The following sera were used. Convalescent sera from fully vaccinated individuals who had been infected with BA.2 after full vaccination (9 2-dose vaccinated and 5 3-dose vaccinated. 14 donors in total) (**a**), and BA.5 after full vaccination (2 2-dose vaccinated donors, 17 3-dose vaccinated donors and 1 4-dose vaccinated donor. 20 donors in total) (**b**). 4-dose vaccine sera collected at 1 month after the 4-dose monovalent vaccine (15 donors) (**c**), BA.1 bivalent vaccine (20 donors) (**d**), and BA.5 bivalent vaccine (21 donors) (**e**). **f** Sera from hamsters infected with BA.2 (12 hamsters), BA.5 (12 hamsters), BQ.1.1 (6 hamsters), BA.2.75 (12 hamsters), and XBB.1 (6 hamsters).

**g** Antigenic cartography based on the results of neutralization assays using hamster sera (Fig. 2f). Assays for each serum sample were performed in triplicate to determine the 50% neutralization titer (NT$_{50}$). Each dot represents one NT$_{50}$ value, and the geometric mean and 95% confidential interval (CI) are shown. Statistically significant differences were determined by two-sided Wilcoxon signed-rank tests. The $P$ values versus BA.2 (**a**), BA.5 (**b**), or XBB.1 (**c–f**) are indicated in the panels. For the BA.2 derivatives (**a–e**), statistically significant differences ($P < 0.05$) versus BA.2 are indicated with asterisks. Red and blue asterisks, respectively, indicate decreased and increased NT$_{50}$s. The horizontal dashed line indicates the detection limit (120-fold). Information on the convalescent donors is summarized in Supplementary Table 5. Source data are provided with this paper.

with BA.2 derivatives revealed that the Y144del mutation (1.8-fold, $P = 0.016$) resulted in resistance to breakthrough BA.5 infection sera (Fig. 2b). Furthermore, in our previous study, we showed that G446S, a common substitution of BA.2.75 and XBB, conferred immune resistance to breakthrough BA.5 infection sera[10]. Together, these observations suggest that these two mutations (Y144del and G446S) cooperatively contribute to the resistance against humoral immunity induced by breakthrough BA.5 infection. We then assessed the sensitivity of XBB.1 to the 4-dose vaccine sera. As shown in Fig. 2c–e, XBB.1 significantly escaped from monovalent vaccine sera (23.3-fold, $P < 0.0001$), BA.1 bivalent vaccine sera (19-fold, $P < 0.0001$), and BA.5 bivalent vaccine sera (41-fold, $P < 0.0001$) compared with B.1.1. The neutralization assay using BA.2-based derivatives showed that multiple substitutions, especially V445P and N460K, also contributed to escape from humoral immunity elicited by 4-dose vaccination.

To further evaluate the antigenicity of XBB.1 S, we used sera obtained from infected hamsters at 16 days post-infection (d.p.i.). Since the divergence of infection and vaccination histories in humans has grown rapidly, the immune background of human sera cannot be

identical. On the other hand, the serum obtained from laboratory animals infected with a single strain of a virus can be useful for antigenic comparison among emerging variants, as in influenza virus studies[21]. As shown in Fig. 2f, XBB.1 exhibited profound resistance to the sera obtained from hamsters infected with BA.2, BA.5, BQ.1.1, and BA.2.75. Moreover, XBB.1-infected hamster sera exhibited a remarkable antiviral effect against only XBB.1 (Fig. 2f). The cartography based on the neutralization dataset using hamster sera (Fig. 2f) showed that the cross-reactivity of each Omicron subvariant is correlated to their phylogenetic relationship (Fig. 1a). The antigenicity of XBB.1 is distinct from that of the other Omicron subvariants tested (Fig. 2g). These observations suggest that XBB.1 is antigenically different from the other Omicron subvariants, including BQ.1.1, and therefore markedly evades BA.2/5 infection-induced herd immunity in the human population.

**ACE2 binding affinity of XBB.1 S**
We then evaluated the features of XBB.1 S that potentially affect viral infection and replication. A yeast surface display assay[2,10,22,23] showed

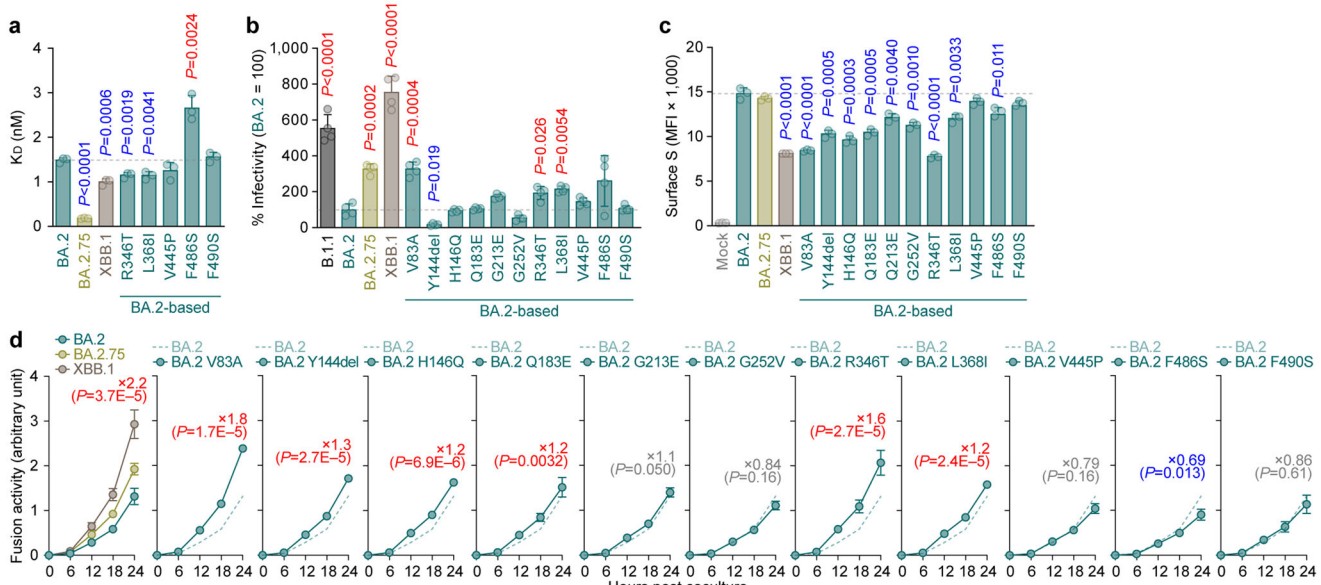

**Fig. 3 | Virological characteristics of XBB.1 in vitro. a** Binding affinity of the receptor binding domain (RBD) of SARS-CoV-2 spike (S) protein to angiotensin-converting enzyme 2 (ACE2) by yeast surface display. The dissociation constant ($K_D$) value indicating the binding affinity of the RBD of the SARS-CoV-2 S protein to soluble ACE2 when expressed on yeast is shown. **b** Pseudovirus assay. HOS-ACE2/TMPRSS2 cells were infected with pseudoviruses bearing each S protein. The amount of input virus was normalized based on the amount of HIV-1 p24 capsid protein. The percent infectivity compared to that of the virus pseudotyped with the BA.2 S protein are shown. **c, d**, S-based fusion assay. **c** S protein expression on the cell surface. The summarized data are shown. **d** S-based fusion assay in Calu-3 cells. The recorded fusion activity (arbitrary units) is shown. The dashed green line indicates the result of BA.2. The red number in each panel indicates the fold difference between BA.2 and the derivative tested (XBB.1 in the top left panel) at 24 h post coculture. Assays were performed in triplicate (**a, c**) or quadruplicate (**b, d**). The presented data are expressed as the average ± standard deviation (SD). In (**a**–**c**), each dot indicates the result of an individual replicate. In (**a**–**d**), the dashed horizontal lines indicate the value of BA.2. In (**a**–**c**), statistically significant differences (\*$P < 0.05$) versus BA.2 were determined by two-sided Student's $t$ tests. Red and blue asterisks, respectively, indicate increased and decreased values. In (**d**), statistically significant differences versus BA.2.75 across timepoints were determined by multiple regression. The familywise error rates (FWERs) calculated using the Holm method are indicated in the figures. Source data are provided with this paper.

that the binding affinity of XBB.1 S RBD to human ACE2 receptor ($1.00 \pm 0.069$) is significantly lower than that of ancestral BA.2 S RBD ($1.49 \pm 0.054$) (Fig. 3a). As described above (Fig. 1a–c), the four RBD substitutions in XBB.1 compared to BA.2, D339H, G446S, N460K and R493Q, are common to BA.2.75 since a part of the RBD of XBB.1 S is derived from the BA.2.75/BM.1 lineage. In our previous studies[2,10], we demonstrated that the N460K substitution augments ACE2 binding affinity. To address whether other substitutions in the XBB.1 S affect the binding affinity of the S RBD to human ACE2, we prepared a repertoire of BA.2 S RBD that possesses an XBB.1-specific substitution compared to BA.2. Consistent with our recent study[2,10], the R346T substitution, which is common in both XBB.1 and BQ.1.1, significantly increased the binding affinity of BA.2 S RBD to human ACE2 (Fig. 3a). Moreover, the L368I substitution augmented ACE2 binding affinity (Fig. 3a). On the other hand, the F486S substitution significantly decreased ACE2 binding affinity (Fig. 3a). Because the F486V substitution also decreased ACE2 binding affinity[5], our data suggest that amino acid substitution at F486 leads to attenuated ACE2 binding affinity. Our results suggested that the enhanced binding affinity of XBB.1 S RBD compared to BA.2 S RBD is attributed to at least three substitutions in the RBD: R346T, L368I and N460K[2,10]. Nevertheless, the $K_D$ value of XBB.1 S RBD was clearly higher than that of BA.2.75 S RBD ($0.18 \pm 0.069$) (Fig. 3a). In our prior study[10], we showed that the D339H substitution contributes to the augmentation of ACE2 binding affinity only when the backbone is BA.2.75 S RBD. Therefore, the profound binding affinity of BA.2.75 S RBD to human ACE2 would be attributed to the conformation that is composed of multiple substitutions in the BA.2.75 S RBD.

We next assessed viral infectivity using pseudoviruses. As shown in Fig. 3b, the infectivity of the XBB.1 pseudovirus was 7.6-fold greater than that of the BA.2 pseudovirus. Consistent with the results of the

yeast surface display assay (Fig. 3a), two substitutions in the RBD, R346T (1.9-fold) and L368I (2.2-fold), significantly increased pseudovirus infectivity (Fig. 3b). Additionally, although two substitutions in the NTD, Y144del (0.18-fold) and G252V (0.54-fold), significantly decreased pseudovirus infectivity, a substitution in the NTD, V83A (3.3-fold), significantly increased pseudovirus infectivity (Fig. 3b). Together, our results suggest that the XBB.1 S augments its infectious potential through multiple substitutions in the RBD (R346T, L368I and N460K) and NTD (V83A).

**Fusogenicity of XBB.1 S**
The fusogenicity of XBB.1 S was measured by the SARS-CoV-2 S-based fusion assay[2,5,10,24–29]. We first assessed the fusogenicity of BA.2.75. Consistent with previous studies[10,18], the BA.2.75 S exhibited higher fusogenicity than BA.2 S (Supplementary Fig. 2a, b). The assay using the BA.2 S derivatives that harbor respective BA.2.75-specific substitutions revealed that only the N460K substitution significantly increased fusogenicity (Supplementary Fig. 2b). We then assessed the fusogenicity of XBB.1 S. As shown in Fig. 3c, the surface expression level of XBB.1 was significantly lower than that of BA.2 and BA.2.75. The S-based fusion assay showed that XBB.1 S is significantly more fusogenic than BA.2 S (2.2-fold) and BA.2.75 S (1.5-fold) (Fig. 3d). To assess the determinant substitutions in XBB.1 S that are responsible for augmented fusogenicity, we used BA.2 S-based derivatives that harbor separate XBB.1-specific substitutions. We revealed that two substitutions, V83A and R346T, significantly increased fusogenicity (Fig. 3d). Together with the experiments focusing on BA.2.75 S (Supplementary Fig. 2b), our results suggest that two substitutions in the RBD (R346T and N460K) and a substitution in the NTD (V83A) contribute to the augmented fusogenicity of XBB.1 S.

## Structural characteristics of XBB.1 S

To gain structural insights into ACE2 receptor recognition and evasion from neutralizing antibodies by XBB.1 S protein, the structures of the XBB.1 S ectodomain alone and the XBB.1 S-ACE2 complex were determined by cryoelectron microscopy (cryo-EM) analysis. The XBB.1 S ectodomain was reconstructed as two closed states (closed-1 and closed-2) at resolutions of 2.50 Å and 2.51 Å, respectively (Fig. 4a, Supplementary Fig. 3, and Supplementary Table 4). Interestingly, the structure of the RBD one-up state, which has been frequently reported in structures of SARS-CoV-2 S protein including BA.2.75, was hardly observed in the structures of XBB.1 S protein. The two closed states observed in the XBB.1 S protein are similar to the closed conformations of the BA.2.75 S protein reported by our group[10] and by Cao et al.[30]. Comparison of closed-1 and closed-2 showed that RBD and SD1 in the S1 subunit rotated at the hinge, and NTD also slightly shifted (Fig. 4a, bottom). There was no major movement in the overall structure of the S2 subunit in the two closed states. However, some differences were observed in both the S1 and S2 subunits in detail. As a major difference between closed-1 and closed-2, the cryo-EM map around the fusion peptide (residue: 828 to 854) was highly disordered in closed-2 but well observable in closed-1 (Fig. 4b and Supplementary Fig. 4a). This suggests that in closed-2, the fusion peptide could not adopt the stable conformation observed in closed-1 due to the movement of the SD1 domain. Next, amino acid residues 969-997 between heptad-repeat 1 adjacent to the RBD and the central helix shifted slightly. In closed-1, S383 of the RBD interacted with the main chain of D985 of heptad-repeat 1 (HR-1), but in closed-2, S383 and K386 of the RBD interacted with the side chain of D985 of HR-1 (Fig. 4b). As a difference in inter-protomer interactions between closed-1 and closed-2, protomer 1 F375 in closed-1 was located in a hydrophobic pocket formed by neighboring protomer 2 V407, V503, and Y508, and stacking interactions between protomer 1 P373 and protomer 2 H505 were also observed, while F375 in closed-2 was located in a hydrophobic pocket within the same protomer formed by V407 and Y508 (Fig. 4b). As S373P and S375F, which are common to the BA.2 lineage, have been reported to form interactions between protomers[31], the XBB.1 S protein is likely to maintain the properties of BA.2 S protein. The cryo-EM map in closed-1 showed a higher resolution, and the RBD was well packed, while that in closed-2 showed a relatively lower resolution in the RBD than in the whole map (Fig. 4a and Supplementary Fig. 3), suggesting that the RBD region is mobile. This further suggests that the loop containing F375 is trapped between protomers for packing in closed-1 but is flexible in closed-2.

The XBB.1 S-ACE2 complex structures were reconstructed by cryo-EM analysis in both RBD one-up and two-up conformations, with resolutions of 3.18 Å and 2.99 Å, respectively (Fig. 4c, Supplementary Fig. 3, and Supplementary Table 4). In both states, S proteins showed the conventional binding mode with ACE2 in the RBD up conformation[32]. Although the XBB.1 S trimer alone did not exhibit any RBD up conformation, it was able to adopt the up conformation at least in the presence of ACE2. To observe the interaction between RBD and ACE2 with better resolution, local refinement was performed on RBD-ACE2 and reconstructed at a resolution of 3.29 Å (Fig. 4c, Supplementary Fig. 3, and Supplementary Table 4). Similar to the RBD-ACE2 complex structures reported for BA.5 or BQ.1.1 bearing F486V[2,5], the F486S substitution changed the side chain to a less bulky hydrophilic residue and lost hydrophobic interactions with the hydrophobic patch consisting of F28, L79, M82, and Y83 in ACE2 (Fig. 4d and Supplementary Fig. 4b). We have reported that F486V contributes to neutralizing antibody evasion by sacrificing affinity for ACE2 in the BA.5 study[5], and a similar result was observed in the F486S substitution in this study (Figs. 2a, c, d and 3a). While the side chain of ACE2 K31 interacted with the backbone oxygen of BQ.1.1 S RBD F490[2], it was located approximately 4.4 Å away from that of XBB.1 S RBD F490S, resulting in interaction with the side chain of XBB.1 S RBD Q493, which

also interacts with ACE2 H34 in an alternative conformation (Fig. 4d and Supplementary Fig. 4b). The F490S substitution was outside the hydrophobic environment formed by T470, I472, P491 and L492 (Fig. 4d and Supplementary Fig. 4b). Similar to the other SARS-CoV-2 variants[2,5,30,33], the residues L368I, R346T, and V445P were not directly involved in the interaction with ACE2 (Fig. 4d and Supplementary Fig. 4b). Interestingly, the XBB.1 S RBD-ACE2 complex was that the N103-linked glycan of ACE2 was positioned towards the active site inside the ACE2 structure (Fig. 4d and Supplementary Fig. 4b, c), not on the outside, as in most SARS-CoV-2 S RBD ACE2 complexes reported thus far by cryo-EM or X-ray crystal structures[32,34]. However, the corresponding glycan in BA.1 S RBD-ACE2[35] and BA.3 S RBD-ACE2[36] structures was located inside the ACE2 structure. In these two structures, branched mannoses were not visible, but in the XBB.1 S RBD-ACE2 structure, branched mannoses were observed on the active site, which consisted of H374, E375, H378 and E402 (Fig. 4d and Supplementary Fig. 4c). The N103-linked glycan of ACE2 interacted with H374, H378, H401, F504, H505, Y510 and R514 in ACE2 (Fig. 4d and Supplementary Fig. 4c).

## Virological characteristics of XBB.1 in vitro

To investigate the growth kinetics of XBB.1 in vitro cell culture systems, we inoculated clinical isolates of BA.2[27], BA.2.75[10], and XBB.1 into multiple cell cultures. The growth kinetics of XBB.1 in Vero cells (Fig. 5a), Calu-3 cells (Fig. 5b), the human airway organoid-derived air-liquid interface (AO-ALI) system (Fig. 5c), and human induced pluripotent stem cell (iPSC)-derived airway epithelial cells (Fig. 5d) were comparable to those of BA.2.75. On the other hand, XBB.1 replicated more efficiently than BA.2.75 in VeroE6/TMPRSS2 cells (Fig. 5e). Similar to our previous study[10], the growth of BA.2.75 was significantly greater than that of BA.2 in human iPSC-derived alveolar epithelial cells (Fig. 5f). However, XBB.1 was less replicative than BA.2.75 in this culture system (Fig. 5f).

To quantitatively assess the impact of XBB.1 infection on the airway epithelial-endothelial barrier, we used an airway-on-a-chip system[2,10,37,38]. By measuring the amount of virus that invaded from the top channel (Fig. 5g, left) to the bottom channel (Fig. 5g, right), we were able to evaluate the ability of viruses to disrupt the airway epithelial-endothelial barriers. Notably, the percentage of virus that invaded the bottom channel of XBB.1-infected airway-on-chips was significantly higher than that of BA.2.75-infected airway-on-chips (Fig. 5h). As shown in Supplementary Fig. 5a, the measured viral RNA load mirrored the amount of infectious viral particles. Together with the findings of the S-based fusion assay (Fig. 3d), these results suggest that XBB.1 is more fusogenic than BA.2.75.

## Virological characteristics of XBB.1 in vivo

To investigate the virological features of XBB.1 in vivo, we inoculated hamster with clinical isolates of Delta[26], BA.2.75[10], and XBB.1. Delta was used as a positive control because Delta is the most pathogenic variant to date[10,25,26,39]. Consistent with our previous studies[2,10,25,26], Delta infection resulted in weight loss (Fig. 6a, left). On the other hand, the body weights of BA.2.75- and XBB.1-infected hamsters were stable and comparable (Fig. 6a, left). We then analysed the pulmonary function of infected hamsters as reflected by two parameters, enhanced pause (Penh) and the ratio of time to peak expiratory flow relative to the total expiratory time (Rpef). Among the four groups, Delta infection resulted in significant differences in these two respiratory parameters compared to XBB.1 (Fig. 6a, middle and right), suggesting that XBB.1 is less pathogenic than Delta. In contrast, although the Penh and Rpef values of XBB.1-infected hamsters were significantly different from those of uninfected hamsters, they were comparable to those of BA.2.75-infected hamsters (Fig. 6a, middle and right). These observations suggest that the pathogenicity of XBB.1 is comparable to that of BA.2.75.

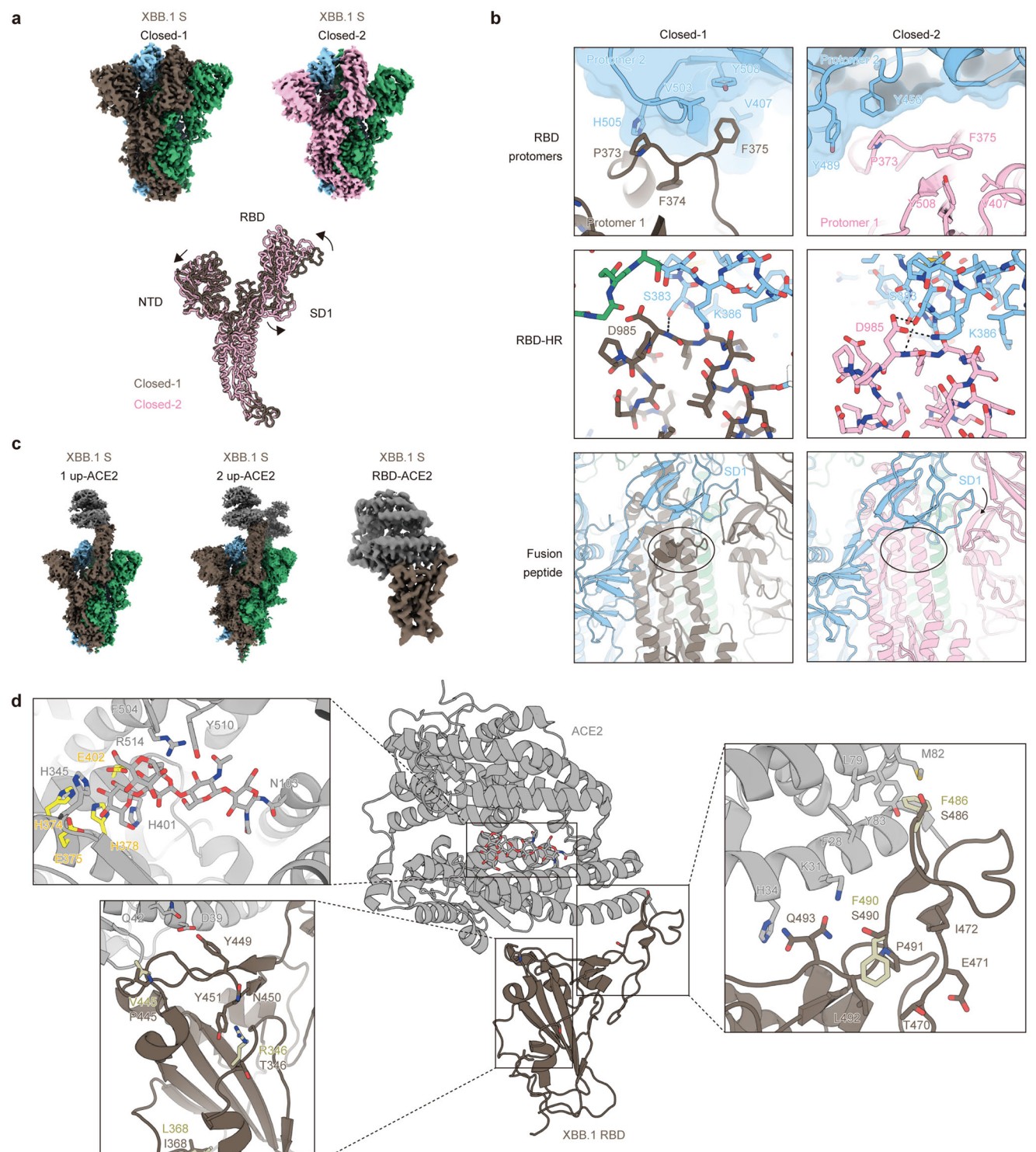

**Fig. 4 | Overall cryo-EM structure of XBB.1 S and ACE2. a** (Top) Cryo-EM maps of XBB.1 spike (S) protein trimer closed-1 state (left) and closed-2 state (right). Each protomer is colored brown, blue, green (closed-1) or pink, blue, green (closed-2). (Bottom) Superimposed structures of XBB.1 S protomers between closed-1 state (brown) and closed-2 state (pink). **b** Close-up and corresponding views of the closed-1 and closed-2 structures (same colors as m). (Top) A loop containing F375 at the protomer interface in the receptor binding domain (RBD) region. Each adjacent protomer is shown with the surface model (transparent, blue). (Middle) Interfaces between RBD and heptad repeat-1 (HR-1). Dashed lines represent hydrogen bonds. (Bottom) Structural difference around fusion peptides (shown in cartoon)

surrounded by a circle. **c** Cryo-EM maps of XBB.1 S protein (same colors as m) bound to angiotensin-converting enzyme 2 (ACE2) (gray) in one-up state (left), two-up state (middle), or RBD-ACE2 interface (right). **d** Structure of RBD-ACE2 complex (same colors as o). In close-up views, corresponding five residues in the BA.2.75 RBD-ACE2 complex structure (PDB: 8ASY)[96] different from that of XBB.1 (brown stick) are shown in pastel yellow sticks. Residues interacting with these five amino acid residues in the XBB.1 or the BA.2.75 RBD, as well as residues recognizing the N103-linked glycan of ACE2, are represented by stick models. Residues of the HEXXH motif in the active site of ACE2 are highlighted in yellow.

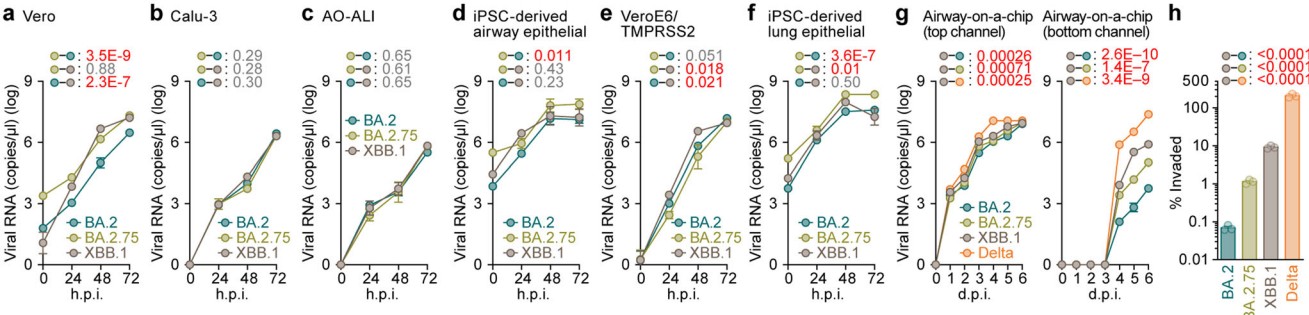

**Fig. 5 | Growth kinetics of XBB.1.** Clinical isolates of BA.2, BA.2.75, XBB.1 and Delta (only in **g**, **h**) were inoculated into Vero cells (**a**), Calu-3 cells (**b**), the human airway organoid-derived air-liquid interface (AO-ALI) system (**c**), human induced pluripotent stem cell (iPSC)-derived airway epithelial cells (**d**), VeroE6/TMPRSS2 cells (**e**), iPSC-derived lung epithelial cells (**f**) and an airway-on-a-chip system (**g**). The copy numbers of viral RNA in the culture supernatant (**a**, **b**, **e**), the apical sides of cultures (**c**, **d**, **f**), and the top (**g**, left) and bottom (**g**, right) channels of an airway-on-a-chip were routinely quantified by RT–qPCR. In (**h**), the percentage of viral RNA load in the bottom channel per top channel at 6 days post-infection (d.p.i.) (i.e., % invaded virus from the top channel to the bottom channel) is shown. Assays were performed in triplicate (**g**, **h**) or quadruplicate (**a**–**f**). The presented data are expressed as the average ± standard error of mean (SEM). In (**h**), each dot indicates the result of an individual replicate. In (**d**–**k**), statistically significant differences across timepoints were determined by multiple regression. In (**h**), statistically significant differences versus XBB.1 were determined by two-sided Student's $t$ tests. The familywise error rates (FWERs) calculated using the Holm method (**a**–**g**) or $P$ values (**h**) are indicated in the figures. Source data are provided with this paper.

To address viral spread in hamsters, we measured the viral RNA load in oral swabs. Although the viral RNA loads of the hamsters infected with XBB.1 were significantly lower than those infected with Delta, there was no significant difference between XBB.1 and BA.2.75 (Fig. 6b, left). To assess the efficacy of viral spread in the respiratory tissues, we collected the lungs of infected hamsters at 2 and 5 d.p.i. and separated them into the hilum and periphery regions. However, the viral RNA loads in both the lung hilum and the periphery of XBB.1-infected hamsters were significantly lower than those of BA.2.75- and Delta-infected hamsters (Fig. 6b, middle and right), suggesting that XBB.1 spreads less efficiently in the lungs of infected hamsters than BA.2.75 and XBB.1. Similar to the in vitro cell culture experiments (Supplementary Fig. 5a), the viral RNA load accurately reflected the infectious viral titer (Supplementary Fig. 5b). We then investigated viral spread in respiratory tissues by immunohistochemical (IHC) analysis targeting the viral nucleocapsid (N) protein. As shown in Fig. 6c and Supplementary Fig. 6a, the percentage of N-positive cells in the lungs of XBB.1-infected hamsters was significantly lower than that in the lungs of BA.2.75- and Delta-infected hamsters. These data suggest that the spreading efficiency of XBB.1 in the lungs of infected hamsters is comparable to or even lower than that of BA.2.75.

### Intrinsic pathogenicity of XBB.1

To investigate the pathogenicity of XBB.1 in the lung, the formalin-fixed right lungs of infected hamsters were analysed by carefully identifying the four lobules and main bronchus and lobar bronchi sectioning each lobe along with the bronchial branches. Histopathological scoring was performed as described in previous studies[2,5,10,25–27]. Briefly, bronchitis or bronchiolitis, hemorrhage or congestion, alveolar damage with epithelial apoptosis and macrophage infiltration, type II pneumocytes and the area of the hyperplasia of large type II pneumocytes were evaluated by certified pathologists, and the degree of these pathological findings was arbitrarily scored using a four-tiered system: 0 (negative), 1 (weak), 2 (moderate), and 3 (severe) (Fig. 6d) according to the criteria shown in previous studies[2,5,10,25–27]. Similar to our previous studies[2,10,25,26], four out of the five histological parameters as well as the total score of Delta-infected hamsters were significantly greater than those of XBB.1-infected hamsters (Fig. 6e). We compared the histopathological scores of two Omicron subvariants, the scores of type II pneumocytes, the area of hyperplasia of large type II pneumocytes, and the total histology score of XBB.1-infected hamsters were comparable to those of BA.2.75-infected hamsters (Fig. 6e). Altogether, these histopathological analyses suggest that the intrinsic

pathogenicity of XBB.1 is lower than that of Delta and comparable to that of BA.2.75.

### Discussion

Here, we illuminated the evolutionary and epidemic dynamics of XBB variant, a recombinant lineage rapidly spreading around the world. Our phylogenetic analyses suggested that XBB emerged through the recombination of two co-circulating BA.2 lineages, BJ.1 and BM.1.1.1 (a progeny of BA.2.75), during the summer of 2022 (Fig. 1). Furthermore, XBB shows substantially higher $R_e$ than the parental lineages, suggesting that the recombination event increased $R_e$ (i.e., viral fitness). To our knowledge, this is the first documented example of a SARS-CoV-2 variant increasing its fitness through recombination rather than substitutions. Furthermore, we showed that the $R_e$ values of XBB lineages are comparable with or slightly higher than those of BQ.1 lineages, and XBB and BQ.1 lineages are becoming dominants in the Eastern and Western regions of the world, respectively. Such regional differences of circulating variants can be explained by two possibilities. The first possibility is that the regional difference is simply due to the geographical distance or distance in the human transportation network from the emergence places of these lineages. Another possibility is that the regional difference is caused by the situation that the fitness of variants changes depending on regions due to the regional variation of immune status. As of February 2023, XBB lineages, particularly XBB.1.5, spread rapidly also in western countries such as the USA[40,41]. Therefore, the former possibility would be more likely, and this variant will spread rapidly worldwide in the near future.

Compared to BA.5, BA.2.75 and even BQ.1.1[2,42], the most remarkable feature of XBB.1 is the profound resistance to antiviral humoral immunity induced by vaccination or breakthrough infections of prior Omicron subvariants (Fig. 2), consistent with reports from other groups[13,42–45]. In fact, our analyses showed that 10 out of 14 breakthrough BA.2 infection sera and 9 out of 20 breakthrough BA.5 infection sera fail to neutralize XBB.1. The neutralization experiments using single mutants showed that multiple substitutions in the XBB.1 S protein cooperatively contribute to the immune resistance of XBB.1, and particularly, not only the substitutions in the RBD but also at least a mutation in the NTD, Y144del, closely associates with the immune resistant property of XBB.1. The effect of Y144del mutation on immune resistance is reported in a recent study by Cao et al.[13], and this mutation has been observed in previous variants of concern such as Alpha[46] and Omicron BA.1[47]. Furthermore, we previously showed that the Mu variant, one of the previous variants of interest, also has a mutation in the

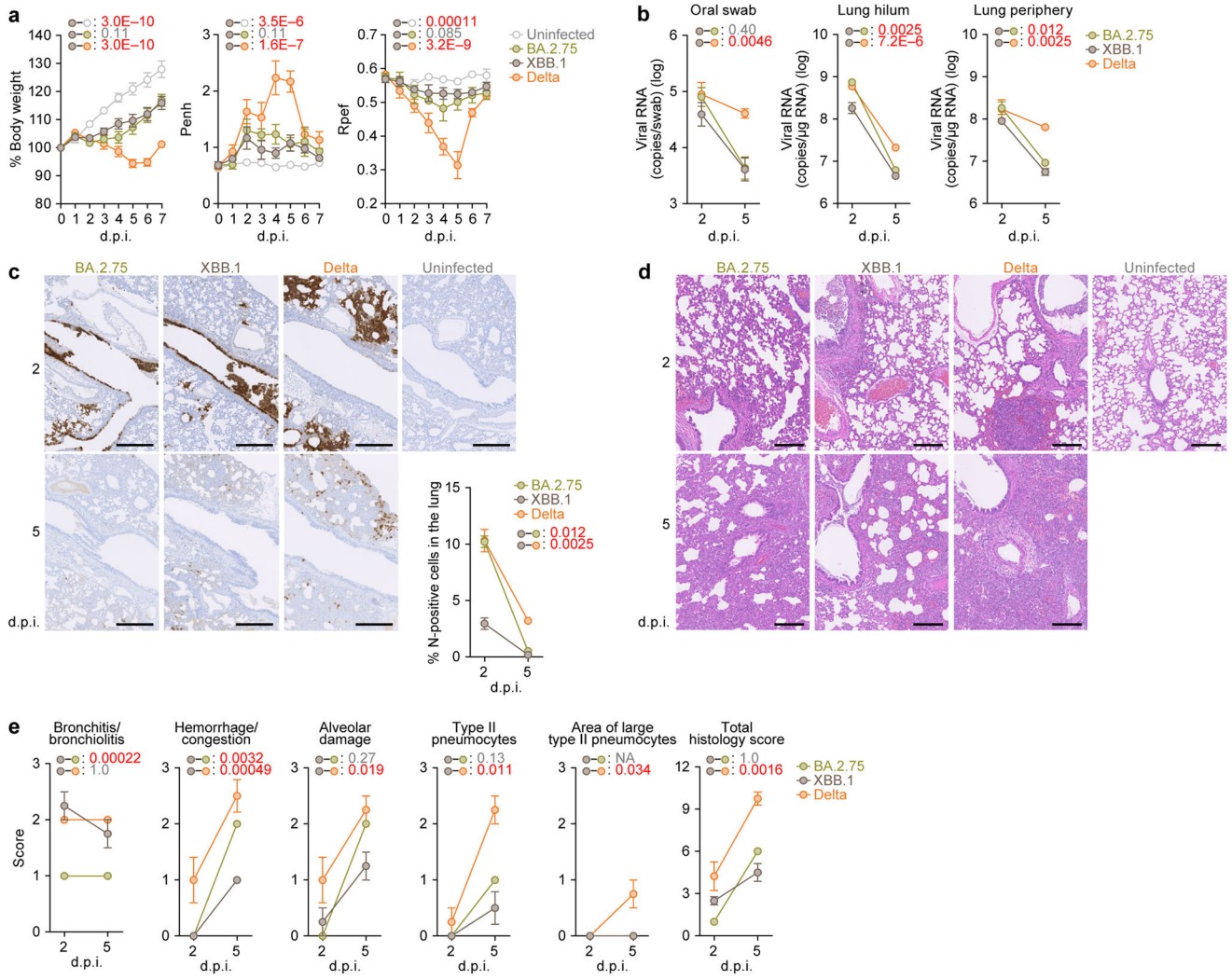

**Fig. 6 | Virological characteristics of XBB.1 in vivo.** Syrian hamsters were intranasally inoculated with BA.2.75, XBB.1 and Delta. Six hamsters of the same age were intranasally inoculated with saline (uninfected). Six hamsters per group were used to routinely measure the respective parameters (**a**). Four hamsters per group were euthanized at 2 and 5 days post-infection (d.p.i.) and used for virological and pathological analysis (**b**–**e**). **a** Body weight, enhanced pause (Penh), and the ratio of time to peak expiratory flow relative to the total expiratory time (Rpef) values of infected hamsters ($n = 6$ per infection group). **b** (Left) Viral RNA loads in the oral swab ($n = 6$ per infection group). (Middle and right) Viral RNA loads in the lung hilum (middle) and lung periphery (right) of infected hamsters ($n = 4$ per infection group). **c** Immunohistochemical (IHC) analysis of the viral nucleocapsid (N) protein in the lungs at 2 d.p.i. (top) and 5 d.p.i. (bottom) of infected hamsters. Representative figures (left, N-positive cells are shown in brown) and the percentage of N-positive cells in whole lung lobes (right, $n = 4$ per infection group) are shown. The raw data are shown in Supplementary Fig. 6. **d**, **e**, Hematoxylin and eosin (H&E) staining of the lungs of infected hamsters. Representative figures are shown in (**d**). Uninfected lung alveolar space and bronchioles are also shown. **e** Histopathological scoring of lung lesions ($n = 4$ per infection group). Representative pathological features are reported in our previous studies[2,5,10,25–27]. In (**a**–**c**), data are presented as the average ± standard error of mean (SEM). In (**a**, **b**, **c**, **e**), statistically significant differences between XBB.1 and other variants across timepoints were determined by multiple regression. In (**a**), the 0 d.p.i. data were excluded from the analyses. The familywise error rates (FWERs) calculated using the Holm method are indicated in the figures. Scale bars, 500 μm (**c**); 200 μm (**d**). Source data are provided with this paper.

region including Y144 (i.e., YY144-145TSN), which contributes to the robust immune escape of this variant[48,49]. Additionally, the region including Y144 is proposed that it is one of the major epitopes of NTD targeting neutralization antibodies[50]. Together, these observations suggest that Y144del mutation in XBB.1 S contributed to escape from these NTD targeting neutralization antibodies.

A series of our previous studies[2,5,10] showed that substitutions in S that are associated with escape from humoral immunity tend to lead to the acquisition of substitutions enhancing the ACE2 binding affinity or viral infectivity probably to compensate for the negative effects of the immune escape-associated substitutions on the ACE2 binding affinity. In the present study, we show that XBB harbors both the immune escape-associated substitutions (i.e., Y144del and F486S) and the infectivity-enhancing substitutions (i.e., V83A and N460K). Importantly, XBB emerged through recombination in the *S* gene, and Y144del and V83A are located on the 5′ recombinant fragment while F486S and N460K are on the 3′ fragment. This means that XBB acquired two pairs of a pair of immune escape-associated and infectivity-enhancing substitutions by only one recombination event. Harboring the two sets of the substitution pairs would be one of the causes why XBB shows higher $R_e$ than other Omicron subvariants. Together, although XBB emerged via a unique evolutionary pathway, our data suggest that XBB also follows the same evolutionary rule with other Omicron subvariants.

A notable common feature between the ACE2 complexes with BQ.1.1[2] or XBB.1 is the structural inhibition of the active site of ACE2. In

the BQ.1.1 RBD-ACE2 complex structure, ACE2 showed the same closed form as an inhibitor-bound form[2]. On the other hand, in the XBB.1 RBD-ACE2 complex structure, ACE2 showed an open form but the N103-linked glycan of ACE2 was found to intrude around the active site of ACE2, which was structurally considered to be an inhibitory factor for its active form. The significance of the effects of BQ.1.1 and XBB.1 on the active form of ACE2 remains to be further study, but it is noteworthy that different variants that emerged around the same period are similarly affecting the structural inhibitions in the active site of ACE2. For fusogenicity, based on the XBB.1 S-ACE2 complex structure, enhanced affinity to ACE2 by R346T in XBB.1 would be an indirect effect. Although R346T does not directly interact with ACE2, R346T lost interactions with N450 and Y451 forming a loop, together with Y449 interacting with ACE2, in XBB.1, compared to those of BA.2.75 (Fig. 4d and Supplementary Fig. 4b)[10,30]. Considering the results of the ACE2 binding assay showing that R346T enhanced affinity to ACE2 (Fig. 3a), and the cell-based fusion assay showing that R346T promoted membrane fusion activity (Fig. 3e), changes in the interactions around R346T in the XBB.1 S trimer may indirectly affect the binding to ACE2.

In our previous studies focusing on some Omicron subvariants such as BA.1[25], BA.2[27], BA.5[5], and BA.2.75[10], viral fusogenicity in in vitro experiments was well correlated to viral intrinsic pathogenicity in a hamster model. However, although the fusogenicity of XBB.1 was greater than that of BA.2.75, one of the parental lineages of XBB, the intrinsic pathogenicity of XBB.1 was comparable or even lower than that of BA.2.75. The discrepancy between viral fusogenicity and intrinsic pathogenicity was also observed in another Omicron subvariant of concern at the end of 2022, BQ.1.1[2]. The discrepancy between viral fusogenicity and intrinsic pathogenicity may be explained by at least three possibilities. First, certain mutations in the non-S region of the XBB.1 genome can attenuate viral pathogenicity augmented by the higher fusogenicity compared with BA.2.75. There are at least seven substitutions in the non-S region of XBB.1 when compared to that of BA.2.75, and some of these mutations may attenuate the viral intrinsic pathogenicity (Supplementary Fig. 1a). Second, a theoretical study by Sasaki, Lion, and Boots provided a possibility that antigenic escape can augment viral pathogenicity[51]. Since we demonstrated that at least two descendants of BA.2, BA.5[5] and BA.2.75[10], increased their intrinsic pathogenicity, this theory may explain the evolution of Omicron. More importantly, this theory also predicts that there is a limitation to increase viral pathogenicity[51]. Together with our observations, it might be possible to assume that the pathogenicity of the Omicron lineage has already reached a plateau. Third, in the cases of BQ.1.1[2] and XBB.1, it might be possible that the tropism and affinity of S proteins of these variants are different between human ACE2 and hamster ACE2, and therefore, a hamster model may not reproduce the human condition.

In summary, our results suggest that XBB shows higher fitness and is resistant to the antiviral humoral immunity induced by breakthrough infections of prior Omicron variants. Although various "local variants" including XBB have simultaneously and convergently emerged in late 2022, local variants showing a higher fitness will eventually spread to the whole world, like XBB. Therefore, continued in-depth viral genomic surveillance and real-time evaluation of the risk of newly emerging SARS-CoV-2 variants, even though considered local variants at the time of emergence, should be crucial.

## Methods
### Ethics statement
All experiments with hamsters were performed in accordance with the Science Council of Japan's Guidelines for the Proper Conduct of Animal Experiments. The protocols were approved by the Institutional Animal Care and Use Committee of National University Corporation Hokkaido University (approval ID: 20-0123, 23-0003 and 20-0060). All protocols involving specimens from human subjects recruited at Interpark Kuramochi Clinic was reviewed and approved by the Institutional Review Board of Interpark Kuramochi Clinic (approval ID: G2021-004). We have obtained consent to publish information that identifies individuals (including three or more indirect identifiers such as exact age, sex, medical history, vaccination history or medical center of the study participants). All human subjects provided written informed consent. All protocols for the use of human specimens were reviewed and approved by the Institutional Review Boards of The Institute of Medical Science, The University of Tokyo (approval IDs: 2021-1-0416 and 2021-18-0617) and University of Miyazaki (approval ID: O-1021).

### Human serum collection
Convalescent sera were collected from fully vaccinated individuals who had been infected with BA.2 (9 2-dose vaccinated and 5 3-dose vaccinated; 11–61 days after testing. $n = 14$ in total; average age: 47 years, range: 24–84 years, 64% male) (Fig. 2a), and fully vaccinated individuals who had been infected with BA.5 (2 2-dose vaccinated, 17 3-dose vaccinated and 1 4-dose vaccinated; 10–23 days after testing. $n = 20$ in total; average age: 51 years, range: 25–73 years, 45% male) (Fig. 2b). The SARS-CoV-2 variants were identified as previously described[5,10,27]. 4-dose vaccine sera from individuals who had been vaccinated with the monovalent vaccine (15 donors; average age: 42 years, range: 30–56 years, 40% male) (Fig. 2c), BA.1 bivalent vaccine (20 donors; average age: 55 years, range: 30–80 years, 35% male) (Fig. 2d), and BA.5 bivalent vaccine (21 donors; average age: 51 years, range: 18–86 years, 48% male) (Fig. 2e). Sera were inactivated at 56 °C for 30 min and stored at −80 °C until use. The details of the convalescent sera are summarized in Supplementary Table 5.

### Cell culture
HEK293T cells (a human embryonic kidney cell line; ATCC, CRL-3216), HEK293 cells (a human embryonic kidney cell line; ATCC, CRL-1573) and HOS-ACE2/TMPRSS2 cells (HOS cells stably expressing human ACE2 and TMPRSS2)[52,53] were maintained in DMEM (high glucose) (Sigma-Aldrich, Cat# 6429-500 ML) containing 10% fetal bovine serum (FBS, Sigma-Aldrich Cat# 172012-500 ML) and 1% penicillin–streptomycin (PS) (Sigma-Aldrich, Cat# P4333-100ML). 293S GnTI(-) cells (HEK293S cells lacking N-acetylglucosaminyltransferase)[54] were maintained in DMEM (Nacalai Tesque, #08458-16) containing 2% FBS without PS. Vero cells [an African green monkey (*Chlorocebus sabaeus*) kidney cell line; JCRB Cell Bank, JCRB0111] were maintained in Eagle's minimum essential medium (EMEM) (Sigma-Aldrich, Cat# M4655-500ML) containing 10% FBS and 1% PS. VeroE6/TMPRSS2 cells (VeroE6 cells stably expressing human TMPRSS2; JCRB Cell Bank, JCRB1819)[55] were maintained in DMEM (low glucose) (Wako, Cat# 041-29775) containing 10% FBS, G418 (1 mg/ml; Nacalai Tesque, Cat# G8168-10ML) and 1% PS. Calu-3 cells (ATCC, HTB-55) were maintained in Eagle's minimum essential medium (EMEM) (Sigma-Aldrich, Cat# M4655-500ML) containing 10% FBS and 1% PS. Calu-3/DSP$_{1-7}$ cells (Calu-3 cells stably expressing DSP$_{1-7}$)[56] were maintained in EMEM (Wako, Cat# 056-08385) containing 20% FBS and 1% PS. Human airway and lung epithelial cells derived from human induced pluripotent stem cells (iPSCs) were manufactured according to established protocols as described below (see "Preparation of human airway and lung epithelial cells from human iPSCs" section) and provided by HiLung Inc. AO-ALI model was generated according to established protocols as described below (see "AO-ALI model" section).

### Viral genome sequencing
Viral genome sequencing was performed as previously described[5]. Briefly, the virus sequences were verified by viral RNA-sequencing analysis. Viral RNA was extracted using a QIAamp viral RNA mini kit

(Qiagen, Cat# 52906). The sequencing library employed for total RNA sequencing was prepared using the NEBNext Ultra RNA Library Prep Kit for Illumina (New England Biolabs, Cat# E7530). Paired-end 76-bp sequencing was performed using a MiSeq system (Illumina) with MiSeq reagent kit v3 (Illumina, Cat# MS-102-3001). Sequencing reads were trimmed using fastp v0.21.0[57] and subsequently mapped to the viral genome sequences of a lineage B isolate (strain Wuhan-Hu-1; GenBank accession number: NC_045512.2)[55] using BWA-MEM v0.7.17[58]. Variant calling, filtering, and annotation were performed using SAMtools v1.9[59] and snpEff v5.0e[60].

## Recombination analysis

As of October 3, 2022, we retrieved a total of 562 sequences satisfying the following criteria from the GISAID database (https://gisaid.org/): (i) human hosts, (ii) collected after 2022, (iii) with length greater than 28,000 base pairs, and (iv) with PANGO lineage designation BJ.1, BM.1, XBB and all their descendants. To ensure that PANGO lineage definitions in our dataset's metadata included the latest circulating lineages, the GISAID metadata were downloaded again on October 15, 2022, and the PANGO lineages of our sequences were updated accordingly. Sequences were aligned to the reference Wuhan-Hu-1 genome (GenBank Accession no. NC_045512.2) and then converted to a multiple sequence alignment using the 'global_profile_alignment.sh' script from the SARS-CoV-2 global phylogeny pipeline[61], utilizing MAFFT[62]. A number of recombination detection methods were performed on the resulting alignment using the Recombination Detection Program (RDP) v.5.21[19], specifically: RDP[63], GENECONV[64], Chimaera[65], MaxChi[66], 3seq[67], BootScan[68] and SiSan[69]. Sequences were assumed to be linear, only recombination events detected consistently by more than 3 independent methods were retrieved and potential false positives were excluded from the final output of RDP5.

## Phylogenetic analyses

To reconstruct the overall relatedness of the XBB parent lineages BJ.1 and BM.1.1.1 to the other Omicron variants (Fig. 1a) we retrieved 100 random sequences from each Omicron PANGO lineages: BA.1, BA.2, BA.4 and BA.5 and 20 random sequences from each younger lineage: BQ.1.1, BA.2.75, BJ.1, and BM.1.1.1. Sequence EPI_ISL_466615 was also added as an outgroup, representing the oldest isolate of B.1.1 obtained in the UK. The sequences were aligned to the reference Wuhan-Hu-1 genome (NC_045512.2) and then converted to a multiple sequence alignment using the 'global_profile_alignment.sh' script from the SARS-CoV-2 global phylogeny pipeline[61] utilizing MAFFT[62]. Fasttree v.2.1[70] was used to infer the phylogeny for the nucleotide alignment under a GTR substitution model (option -gtr).

For inferring the phylogenies of each non-recombinant segment of the XBB variant, we first split the alignment used for the recombination analysis above at genome position 22,920 (the breakpoint inferred by RDP5). Due to the lack of many informative sites of the 3′ end shorter non-recombinant alignment, two quality filtering steps were implemented: (i) the 3′ end of the alignment was trimmed up to the position where none of the sequences had 3′ end gaps and (ii) all sequences with Ns were removed, leading to a reduced alignment of 370 sequences. BA.2 sequence EPI_ISL_10926749 was added to the alignments as an outgroup. Iqtree2 v2.1.3[71] was used for making a phylogenetic for each non-recombinant alignment. The TIM2 + F + I substitution model was used for both trees as selected by the '-m TEST' of iqtree and node support was assessed by performing 1000 ultrafast bootstrap replicates.

Both phylogenies were manually inspected for the presence of temporal signal using TempEst v1.5.3[72]. The 3′ end non-recombinant segment's phylogeny did not have enough substitutions for a root-to-tip regression to be inferred, hence we proceeded with tip-dating analysis only for the 5′ end, longer segment. We used BEAST v.1.10.4[73] to infer a time-calibrated Bayesian phylogeny of this genome segment. To avoid missing information affecting the inference we also removed all sequences containing Ns from the alignment, leading to a reduced dataset of 247 sequences. We used a strict molecular clock model with an exponential growth coalescent prior [https://doi.org/10.1098/rstb.1994.0079]. The HKY substitution model was used, accounting for site heterogeneity with an invariant site and four category Γ distribution model. A clock rate prior with mean of $1 \times 10^{-3}$ and standard deviation of $1 \times 10^{-4}$ was provided – consistent with the accepted rate for SARS-CoV-2[74]–and all XBB sequences were assumed to be monophyletic. Duplicate MCMC chains were run for 100,000,000 states each, sampling every 10,000 states. Convergence was assessed using Tracer v1.7.1 [https://academic.oup.com/sysbio/article/67/5/901/4989127] and maximum clade credibility (MCC) trees were summarized by combining the two chains after removing a 10% burn-in using Log-Combiner (https://beast.community/logcombiner) and TreeAnnotator (https://beast.community/treeannotator).

## Epidemic dynamics analyses

We modeled the epidemic dynamics of viral lineages based on the viral genomic surveillance data deposited in the GISAID database (https://www.gisaid.org/). In the present study, we performed three types of analyses: (i) The estimation of the relative $R_e$ for lineages related to XBB in India (shown in Fig. 1e and Supplementary Fig. 1b), (ii) The estimation of the epidemic frequencies of XBB and BQ.1 lineages in each country as of November 15, 2022 (shown in Fig. 1f), and (iii) The estimation of the global and country-specific $R_e$ value of XBB and BQ.1 lineages in the countries where these variants circulated (Fig. 1g, h and Supplementary Fig. 1c). For the three analyses, the metadata of viral sequences downloaded from the GISAID database on December 1, 2022 was used. We excluded the sequence records with the following features: (i) a lack of collection date (information); (ii) sampling in animals other than humans; (iii) sampling by quarantine; or (iv) without the PANGO lineage information.

To estimate the relative $R_e$ for lineages related to XBB in India, we analyzed the records for samples from India from June 1, 2022 to November 15, 2022. We removed records with >5% undetermined (N) nucleotide sequences from the dataset. We first simplified the viral lineage classification based on the PANGO lineage. We renamed the sublineages of BA.5 as BA.5, and subsequently, we removed the BA.5 sequences harboring any of the convergent S substitutions, S:R346X, S:K444X, and S:N460X from our dataset in order to exclude the sequences belonging to the recent BA.5 sublineages exhibiting particularly higher $R_e$ such as BQ.1.1[2]. Also, we removed the sequences of BA.2.75 harboring any of the convergent S substitutions, S:R346X, S:K444X, S:N460X, and S:F486X. Furthermore, since a part of BA.2.10.1 sequences harbor XBB-characteristic substitutions (S:V83A, S:F486S, and S:F490S) probably due to the misclassification of XBB, we removed the sequences of BA.2.10.1 harboring these XBB-characteristic substitutions. According to the modified viral lineages, we extracted records for viral lineages of interest: BA.2, BA.5, BA.2.75, BM.1, BM1.1, BM.1.1.1, BA.2.10, BJ.1, XBB, and XBB.1. Subsequently, we counted the daily frequency of each viral lineage. Relative $R_e$ value for each viral lineage was estimated according to the Bayesian multinomial logistic model, described in our previous study[5]. Briefly, we estimated the logistic slope parameter $\beta_l$ for each viral lineage using the model and then calculated relative $R_e$ for each lineage $r_l$ as $r_l = \exp(\gamma\beta_l)$ where $\gamma$ is the average viral generation time (2.1 days) (http://sonorouschocolate.com/covid19/index.php?title=Estimating_Generation_Time_Of_Omicron). Parameter estimation was performed via the MCMC approach implemented in CmdStan v2.30.1 (https://mc-stan.org) with CmdStanr v0.5.3 (https://mc-stan.org/cmdstanr/). Four independent MCMC chains were run with 500 and 1,000 steps in the warmup and sampling iterations, respectively. We confirmed that all estimated parameters showed <1.01 R-hat convergence diagnostic values and >200 effective sampling size values, indicating that the

MCMC runs were successfully convergent. Information on the estimated parameters is summarized in Supplementary Table 1.

To estimate the epidemic frequencies of XBB and BQ.1 lineages in each country as of November 15, 2022, we analyzed the records for viral samples collected from August 1, 2022 to November 15, 2022. In data for each country, we counted the daily lineage frequency of BQ.1 (including its decedent sublineages), XBB (including its decedent sublineages), and the other SARS-CoV-2 lineages (referred to as "Other lineages"). We analyzed the data only for countries with a total of ≥1000 samples or ≥50 samples of either the BQ.1 or XBB lineages. In this criterion, 56 countries remained. Subsequently, we fitted the multinomial logistic model described in the paragraph above to the daily lineage frequency data of each country separately, and the epidemic frequency of each viral lineage as of November 15, 2022 in each country was estimated. If the data for November 15, 2022 in a particular country are not available, the lineage frequencies at the latest date in the country were used instead. The estimated lineage frequencies for BQ.1 and XBB in each country were shown on the global map using The R library maps v3.4.1 (https://cran.r-project.org/web/packages/maps/index.html). Information on the estimated lineage frequencies is summarized in Supplementary Table 2.

To estimate the global average and country-specific $R_e$ values for BQ.1 and XBB lineages, we analyzed the sequence records for viral samples collected from August 1, 2022 to November 15, 2022. We defined the sequences of BQ.1 (including its sublineages) harboring S:R346T as BQ.1.1 and the other BQ.1 sequences as BQ.1. Similarly, the sequences of XBB (including its sublineages) harboring S:G252V as XBB and the other XBB sequences as XBB. Subsequently, we extracted the sequence records of BQ.1, BQ.1.1, XBB, and XBB.1 in addition to BA.5 (including its sublineages) and BA.2.75 (including its sublineages), which are predominant lineages before the BQ.1 and XBB emergencies. Next, we counted the daily frequency of the lineages above in each country. We analyzed counties with a total of ≥1000 samples and ≥200 samples of either the BQ.1, BQ.1.1, XBB, or XBB.1 lineages. In this criterion, 11 countries (Australia, Austria, Denmark, India, Indonesia, Israel, Malaysia, Peru, Singapore, the UK, and the USA) remained. To estimate the global average $R_e$ values of the lineages above, we employed a hierarchal Bayesian multinomial logistic model, which we established in our previous studies[10,27]. Briefly, this hierarchal model can estimate the global average and country-specific $R_e$ values of lineages of interest simultaneously according to the daily lineage frequency data from multiple countries. The relative $R_e$ of each viral lineage $l$ in each county $s$ ($r_{ls}$) was calculated according to the country-specific slope parameter, $\beta_{ls}$, as $r_{ls} = \exp(\gamma\beta_{ls})$ where $\gamma$ is the average viral generation time (2.1 days). Similarly, the global average relative $R_e$ of each viral lineage was calculated according to the global average slope parameter, $\beta_l$, as $r_l = \exp(\gamma\beta_l)$. For parameter estimation, the global average intercept and slope parameters of the BA.5 variant were fixed at 0. Consequently, the relative $R_e$ of BA.5 was fixed at 1, and those of the other lineages were estimated relative to that of BA.5. Parameter estimation was performed via the MCMC approach implemented in CmdStan v2.30.1 (https://mc-stan.org) with CmdStanr v0.5.3 (https://mc-stan.org/cmdstanr/). Four independent MCMC chains were run with 500 and 2,000 steps in the warmup and sampling iterations, respectively. We confirmed that all estimated parameters showed <1.01 R-hat convergence diagnostic values and >200 effective sampling size values, indicating that the MCMC runs were successfully convergent. Information on the estimated parameters is summarized in Supplementary Table 3.

## Plasmid construction
Plasmids expressing the codon-optimized SARS-CoV-2 S proteins of B.1.1 (the parental D614G-bearing variant), BA.2 and BA.5, BQ.1.1 and BA.2.75 were prepared in our previous studies[2,5,10,24,27,75]. Plasmids expressing the codon-optimized S proteins of XBB.1 and BA.2 S-based

derivatives were generated by site-directed overlap extension PCR using the primers listed in Supplementary Table 6. The resulting PCR fragment was digested with KpnI (New England Biolabs, Cat# R0142S) and NotI (New England Biolabs, Cat# R1089S) and inserted into the corresponding site of the pCAGGS vector[76]. Nucleotide sequences were determined by DNA sequencing services (Eurofins), and the sequence data were analyzed by Sequencher v5.1 software (Gene Codes Corporation).

## Neutralization assay
Pseudoviruses were prepared as previously described[2,5,10,26,27,29,48,49,53,56,75,77]. Briefly, lentivirus (HIV-1)-based, luciferase-expressing reporter viruses were pseudotyped with SARS-CoV-2 S proteins. HEK293T cells (1,000,000 cells) were cotransfected with 1 μg psPAX2-IN/HiBiT[52], 1 μg pWPI-Luc2[52], and 500 ng plasmids expressing parental S or its derivatives using PEI Max (Polysciences, Cat# 24765-1) according to the manufacturer's protocol. Two days posttransfection, the culture supernatants were harvested and centrifuged. The pseudoviruses were stored at −80 °C until use.

The neutralization assay (Fig. 2) was prepared as previously described[2,5,10,26,27,29,48,49,53,56,75,77]. Briefly, the SARS-CoV-2 S pseudoviruses (counting ~20,000 relative light units) were incubated with serially diluted (120-fold to 87,480-fold dilution at the final concentration) heat-inactivated sera at 37 °C for 1 h. Pseudoviruses without sera were included as controls. Then, a 40 μl mixture of pseudovirus and serum/antibody was added to HOS-ACE2/TMPRSS2 cells (10,000 cells/50 μl) in a 96-well white plate. At 2 d.p.i., the infected cells were lysed with a One-Glo luciferase assay system (Promega, Cat# E6130), a Bright-Glo luciferase assay system (Promega, Cat# E2650), or a britelite plus Reporter Gene Assay System (PerkinElmer, Cat# 6111 6066769), and the luminescent signal was measured using a GloMax explorer multimode microplate reader 3500 (Promega) or Centro XS3 LB960 (Berthold Technologies). The assay of each serum sample was performed in triplicate, and the 50% neutralization titer ($NT_{50}$) was calculated using Prism 9 software v9.1.1 (GraphPad Software).

## SARS-CoV-2 preparation and titration
The working virus stocks of SARS-CoV-2 were prepared and titrated as previously described[2,5,10,24–27,29,38,78]. In this study, clinical isolates of B.1.1 (strain TKYE610670; GISAID ID: EPI_ISL_479681)[25], Delta (B.1.617.2, strain TKYTK1734; GISAID ID: EPI_ISL_2378732)[26], BA.2 (strain TY40-385; GISAID ID: EPI_ISL_9595859)[5], BA.5 (strain TKYS14631; GISAID ID: EPI_ISL_12812500)[10,38], BA.2.75 (strain TY41-716; GISAID ID: EPI_ISL_13969765)[10] and XBB.1 (strain TY41-795; GISAID ID: EPI_ISL_15669344) were used. In brief, 20 μl of the seed virus was inoculated into VeroE6/TMPRSS2 cells (5,000,000 cells in a T-75 flask). One h.p.i., the culture medium was replaced with DMEM (low glucose) (Wako, Cat# 041-29775) containing 2% FBS and 1% PS. At 3 d.p.i., the culture medium was harvested and centrifuged, and the supernatants were collected as the working virus stock.

The titer of the prepared working virus was measured as the 50% tissue culture infectious dose ($TCID_{50}$). Briefly, one day before infection, VeroE6/TMPRSS2 cells (10,000 cells) were seeded into a 96-well plate. Serially diluted virus stocks were inoculated into the cells and incubated at 37 °C for 4 days. The cells were observed under a microscope to judge the CPE appearance. The value of $TCID_{50}$/ml was calculated with the Reed–Muench method[79].

For verification of the sequences of SARS-CoV-2 working viruses, viral RNA was extracted from the working viruses using a QIAamp viral RNA mini kit (Qiagen, Cat# 52906) and viral genome sequences were analyzed as described above (see "Viral genome sequencing" section). Information on the unexpected substitutions detected is summarized in Supplementary Table S7, and the raw data are deposited in the Sequence Read Archive (accession ID: PRJDB14899).

## Yeast surface display

Yeast surface display (Fig. 3a) was performed as previously described[2,10,22,23]. Briefly, the RBD genes ["construct 3" in reference[23], covering residues 330–528] in the pJYDC1 plasmid were cloned by restriction enzyme-free cloning and transformed into the EBY100 Saccharomyces cerevisiae. The primers are listed in Supplementary Table S6. The expression media 1/9[80] was inoculated (OD 1) by overnight (220 rpm, 30 °C, SD-CAA media) grown culture, followed by cultivation for 24 h at 20 °C. The medium was supplemented with 10 mM DMSO solubilized bilirubin (Sigma-Aldrich, Cat# 14370-1 G) for expression cocultivation labeling [pJYDC1, eUnaG2 reporter holo-form formation, green/yellow fluorescence (excitation at 498 nm, emission at 527 nm)]. Cells (100 µl aliquots) were collected by centrifugation (3000 g, 3 min), washed in ice-cold PBSB buffer (PBS with 1 mg/ml BSA), and resuspended in an analysis solution with a series of CF®640 R succinimidyl ester labeled (Biotium, Cat# 92108) ACE2 peptidase domain (residues 18–740) concentrations. The peptidase domain of wild-type ACE2 and ACE2 N90Q were produced and purified as previously described[23]. The reaction volume was adjusted (1–100 ml) to avoid the ligand depletion effect, and the suspension was incubated overnight in a rotator shaker (10 rpm, 4 °C). Incubated samples were washed with PBSB buffer, transferred into a 96-well plate (Thermo Fisher Scientific, Cat# 268200), and analyzed by a CytoFLEX S Flow Cytometer (Beckman Coulter, USA, Cat#. N0-V4-B2-Y4) with the gating strategy described previously[23]. The eUnaG2 signals were compensated by CytExpert software (Beckman Coulter). The mean binding signal (FL4-A) values of RBD-expressing cells, subtracted by signals of nonexpressing populations, were subjected to the determination of the dissociation constant $K_D$, $Y_D$ by a noncooperative Hill equation fitted by nonlinear least-squares regression using Python v3.7 fitted together with two additional parameters describing the titration curve[23].

## Pseudovirus infection

Pseudovirus infection (Fig. 3b) was performed as previously described[2,5,10,26,27,29,48,49,53,56,75,77]. Briefly, the amount of pseudoviruses prepared was quantified by the HiBiT assay using a Nano Glo HiBiT lytic detection system (Promega, Cat# N3040) as previously described[52,81]. For measurement of pseudovirus infectivity, the same amount of pseudoviruses (normalized to the HiBiT value, which indicates the amount of HIV-1 p24 antigen) was inoculated into HOS-ACE2/TMPRSS2 cells and viral infectivity was measured as described above (see "Neutralization assay" section).

## SARS-CoV-2 S-based fusion assay

A SARS-CoV-2 S-based fusion assay (Fig. 3c, d and Supplementary Fig. 2) was performed as previously described[2,5,10,24–29]. Briefly, on day 1, effector cells (i.e., S-expressing cells) and target cells (Calu-3/DSP$_{1-7}$ cells) were prepared at a density of 0.6–0.8 × 10$^6$ cells in a 6-well plate. On day 2, for the preparation of effector cells, HEK293 cells were cotransfected with the S expression plasmids (400 ng) and pDSP$_{8-11}$ (reference[82]) (400 ng) using TransIT-LT1 (Takara, Cat# MIR2300). On day 3 (24 h posttransfection), 16,000 effector cells were detached and reseeded into a 96-well black plate (PerkinElmer, Cat# 6005225), and target cells were reseeded at a density of 1,000,000 cells/2 ml/well in 6-well plates. On day 4 (48 h posttransfection), target cells were incubated with EnduRen live cell substrate (Promega, Cat# E6481) for 3 h and then detached, and 32,000 target cells were added to a 96-well plate with effector cells. Renilla luciferase activity was measured at the indicated time points using Centro XS3 LB960 (Berthold Technologies). For measurement of the surface expression level of the S protein, effector cells were stained with rabbit anti-SARS-CoV-2 S S1/S2 polyclonal antibody (Thermo Fisher Scientific, Cat# PA5-112048, 1:100). Normal rabbit IgG (Southern Biotech, Cat# 0111-01, 1:100) was used as a negative control, and APC-conjugated goat anti-rabbit IgG polyclonal

antibody (Jackson ImmunoResearch, Cat# 111-136-144, 1:50) was used as a secondary antibody. The surface expression level of S proteins (Fig. 3c and Supplementary Fig. 2a) was measured using a FACS Canto II (BD Biosciences) and the data were analyzed using FlowJo software v10.7.1 (BD Biosciences). Gating strategy for flow cytometry is shown in Supplementary Fig. 7. For calculation of fusion activity, Renilla luciferase activity was normalized to the mean fluorescence intensity (MFI) of surface S proteins. The normalized value (i.e., Renilla luciferase activity per the surface S MFI) is shown as fusion activity.

## Protein expression and purification for cryo-EM

Protein expression and purification of XBB.1 S protein ectodomain and human ACE2 were performed as previously described[10]. Briefly, the expression plasmid, pHLsec, encoding the XBB.1 S protein ectodomain bearing six proline substitutions (F817P, A892P, A899P, A942P, K986P and V987P)[83] and the deletion of the furin cleavage site (i.e., RRAR to GSAG substitution) with a T4-foldon domain or soluble human ACE2 ectodomain were transfected into HEK293S GnTI(-) cells. Expressed proteins in the cell-culture supernatant were purified with a cOmplete His-Tag Purification Resin (Roche, Cat# 5893682001) affinity column, followed by Superose 6 Increase 10/300 GL size-exclusion chromatography (Cytiva, Cat# 29091596) with calcium- and magnesium-free PBS buffer.

## Cryo-EM sample preparation and data collection

The solution of XBB.1 S protein was incubated at 37 °C for 1 h before cryo-EM grid preparation. The purified ACE2 was incubated with XBB.1 S protein at a molar ratio of 1:3.2 (spike:ACE2) at 18 °C for 10 min. The samples were then applied to a Quantifoil R2.0/2.0 Cu 300 mesh grid (Quantifoil Micro Tools GmbH), which had been freshly glow-discharged for 60 s at 10 mA using PIB-10 (Vacuum Device). The samples were plunged into liquid ethane using a Vitrobot mark IV (Thermo Fisher Scientific) with the following settings: temperature 18 °C, humidity 100%, blotting time 5 seconds, and blotting force 5.

Movies were collected on a Krios G4 (Thermo Fisher Scientific) operated at 300 kV with a K3 direct electron detector (Gatan) at a nominal magnification of 130,000 (0.67 per physical pixel), using a GIF-Biocontinuum energy filter (Gatan) with a 20 eV slit width. Each micrograph was collected with a total exposure of 1.5 s and a total dose of 50.4 or 56.4 e/Å$^2$ over 50 frames. A total of 3,412 (dataset 1) and 2,900 (dataset 2) movies for XBB.1 S, and A total of 3,630 (dataset 1) and 3,772 (dataset 2) movies for XBB.1 S-ACE2 complexes were collected at a nominal defocus range of 0.8–1.8 µm using EPU software (Thermo Fisher Scientific).

## Cryo-EM image processing

All datasets were processed in cryoSPARC v4.1.2[84]. For XBB.1 S protein trimer alone, movie frames were aligned, dose-weighted, and CTF-estimated using Patch Motion correction and Patch CTF. 852,470 (dataset 1) and 1,114,336 (dataset 2) particles were blob-picked and reference-free 2D classification (K = 150, batch = 200, Iteration = 30) was performed on each dataset separately to remove junk particles. 380,752 particles from two datasets were used for initial model reconstruction and heterogeneous refinement. Two classes of closed states (closed-1 and closed-2) with different RBD orientations were separated in heterogeneous refinement. Closed-1 state was processed by non-uniform refinement with C3 symmetry and CTF refinement to generate the final maps. Since the density of the RBD was unclear for closed-2 state, once the particles were aligned with non-uniform refinement, aligned particles were symmetry-expanded under C3 symmetry operation. 3D classification (K = 4, force hard classification, input mode = simple) focused on the RBD without alignment was performed, and selected classes clearly showed RBD and different conformation with closed-1. A final map of closed-2 state was

 

reconstructed with Non-uniform refinement after removing duplicate particles. To support model building, a local refinement focusing on RBD in closed-2 states was carried out.

For XBB.1 S protein bound to ACE2, the two datasets were pre-processed to 2D classification (K = 150, batch = 200, Iteration = 30), in the same way as XBB.1 S protein trimer alone. The initial model was reconstructed using particles belonging to dataset 1, heterogeneous refinement was performed using all picked 1,630,799 particles. To address the flexibility of the RBD-ACE2 interface, a 3D classification (K = 4, force hard classification, input mode = simple) focused on RBD-ACE2 interface without alignment was performed. A local map of RBD-ACE2 interface was obtained by local refinement using particles belonging to the class with clearly observed ACE2. Since the down RBD was still unclear, a further 3D classification focused on down RBD without alignment was performed to obtain 1-up state global map. Heterogeneous refinement was performed using the remaining particles to obtain 2-up state global map.

The reported global resolutions are based on the gold-standard Fourier shell correlation curves (FSC = 0.143) criterion. Local resolutions were calculated with cryoSPARC[85]. Workflows of data processing were shown in Supplementary Fig. 3. Figures related to data processing and reconstructed maps were prepared with UCSF Chimera v1.15[86] and UCSF Chimera X v1.4[87].

### Cryo-EM model building and analysis

Structures of SARS-CoV-2 BA.2.75 S protein (PDB: 8GS6[10]) and/or human ACE2 protein (PDB:7XBO[36]) were fitted to the corresponding maps using UCSF Chimera. Iterative rounds of manual fitting in Coot v0.9.6[88] and real-space refinement in Phenix v1.20[89] were carried out to improve non-ideal rotamers, bond angles, and Ramachandran outliers. The final model was validated with MolProbity[90]. The structure models shown in surface, cartoon and stick presentation in figures were prepared with PyMOL v2.3.3 (http://pymol.sourceforge.net).

### AO-ALI model

An airway organoid (AO) model was generated according to our previous report[2,10,37,38]. Briefly, normal human bronchial epithelial cells (NHBEs, Cat# CC-2540, Lonza) were used to generate AOs. NHBEs were suspended in 10 mg/ml cold Matrigel growth factor reduced basement membrane matrix (Corning, Cat# 354230). Fifty microliters of cell suspension were solidified on prewarmed cell culture-treated multiple dishes (24-well plates; Thermo Fisher Scientific, Cat# 142475) at 37 °C for 10 min, and then, 500 μl of expansion medium was added to each well. AOs were cultured with AO expansion medium for 10 days. For maturation of the AOs, expanded AOs were cultured with AO differentiation medium for 5 days.

The AO-ALI model (Fig. 5c) was generated according to our previous report[10,91]. For generation of AO-ALI, expanding AOs were dissociated into single cells, and then were seeded into Transwell inserts (Corning, Cat# 3413) in a 24-well plate. AO-ALI was cultured with AO differentiation medium for 5 days to promote their maturation. AO-ALI was infected with SARS-CoV-2 from the apical side.

### Preparation of human airway and lung epithelial cells from human iPSCs

The air-liquid interface culture of airway and lung epithelial cells (Fig. 5d, f) was differentiated from human iPSC-derived lung progenitor cells as previously described[5,10,38,92–94]. Briefly, lung progenitor cells were induced stepwise from human iPSCs according to a 21-day and 4-step protocol[92]. At day 21, lung progenitor cells were isolated with the specific surface antigen carboxypeptidase M and seeded onto the upper chamber of a 24-well Cell Culture Insert (Falcon, #353104), followed by 28-day and 7-day differentiation of airway and lung epithelial cells, respectively. Alveolar differentiation medium with dexamethasone (Sigma-Aldrich, Cat# D4902), KGF (PeproTech, Cat# 100-

19), 8-Br-cAMP (BIOLOG Life Science Institute, Cat# B007), 3-isobutyl 1-methylxanthine (IBMX) (Fujifilm Wako, Cat# 095-03413), CHIR99021 (Axon Medchem, Cat# 1386), and SB431542 (Fujifilm Wako, Cat# 198-16543) was used for the induction of lung epithelial cells. PneumaCult ALI (STEMCELL Technologies, Cat# ST-05001) with heparin (Nacalai Tesque, Cat# 17513-96) and Y-27632 (LC Laboratories, Cat# Y-5301) hydrocortisone (Sigma-Aldrich, Cat# H0135) was used for induction of airway epithelial cells.

### Airway-on-a-chips

Airway-on-a-chips (Fig. 5g, h) were prepared as previously described[2,10,37,38]. Human lung microvascular endothelial cells (HMVEC-L) were obtained from Lonza (Cat# CC-2527) and cultured with EGM-2-MV medium (Lonza, Cat# CC-3202). For preparation of the airway-on-a-chip, first, the bottom channel of a polydimethylsiloxane (PDMS) device was precoated with fibronectin (3 μg/ml, Sigma-Aldrich, Cat# F1141). The microfluidic device was generated according to our previous report[95]. HMVEC-L cells were suspended at 5,000,000 cells/ml in EGM-2-MV medium. Then, 10 μl of suspension medium was injected into the fibronectin-coated bottom channel of the PDMS device. Then, the PDMS device was turned upside down and incubated. After 1 h, the device was turned over, and the EGM2-MV medium was added into the bottom channel. After 4 days, AOs were dissociated and seeded into the top channel. AOs were generated according to our previous report[91]. AOs were dissociated into single cells and then suspended at 5,000,000 cells/ml in the AO differentiation medium. Ten microliter suspension medium was injected into the top channel. After 1 h, the AO differentiation medium was added to the top channel. In the infection experiments (Fig. 5g, h), the AO differentiation medium containing either BA.2, BA.2.75, XBB.1 or Delta isolate (500 TCID$_{50}$) was inoculated into the top channel. At 2 h.p.i., the top and bottom channels were washed and cultured with AO differentiation and EGM2-MV medium, respectively. The culture supernatants were collected, and viral RNA was quantified using RT−qPCR (see "RT−qPCR" section above).

### Microfluidic device

A microfluidic device was generated according to our previous report[10,95]. Briefly, the microfluidic device consisted of two layers of microchannels separated by a semipermeable membrane. The microchannel layers were fabricated from PDMS using a soft lithographic method. PDMS prepolymer (Dow Corning, Cat# SYLGARD 184) at a base to curing agent ratio of 10:1 was cast against a mold composed of SU-8 2150 (MicroChem, Cat# SU-8 2150) patterns formed on a silicon wafer. The cross-sectional size of the microchannels was 1 mm in width and 330 μm in height. Access holes were punched through the PDMS using a 6-mm biopsy punch (Kai Corporation, Cat# BP-L60K) to introduce solutions into the microchannels. Two PDMS layers were bonded to a PET membrane containing 3.0-μm pores (Falcon, Cat# 353091) using a thin layer of liquid PDMS prepolymer as the mortar. PDMS prepolymer was spin-coated (4000 rpm for 60 sec) onto a glass slide. Subsequently, both the top and bottom channel layers were placed on the glass slide to transfer the thin layer of PDMS prepolymer onto the embossed PDMS surfaces. The membrane was then placed onto the bottom layer and sandwiched with the top layer. The combined layers were left at room temperature for 1 day to remove air bubbles and then placed in an oven at 60 °C overnight to cure the PDMS glue. The PDMS devices were sterilized by placing them under UV light for 1 h before the cell culture.

### SARS-CoV-2 infection

One day before infection, Vero cells (10,000 cells), VeroE6/TMPRSS2 cells (10,000 cells) and Calu-3 cells (10,000 cells) were seeded into a 96-well plate. SARS-CoV-2 [1,000 TCID$_{50}$ for Vero cells (Fig. 5a); 100 TCID$_{50}$ for VeroE6/TMPRSS2 cells (Fig. 5e) and Calu-3 cells (Fig. 5b)] was inoculated and incubated at 37 °C for

1 h. The infected cells were washed, and 180 μl of culture medium was added. The culture supernatant (10 μl) was harvested at the indicated timepoints and used for RT–qPCR to quantify the viral RNA copy number (see "RT–qPCR" section below). In the infection experiments using AO-ALI (Fig. 5c), human iPSC-derived airway and lung epithelial cells (Fig. 5d, f), working viruses were diluted with Opti-MEM (Thermo Fisher Scientific, Cat# 11058021). The diluted viruses (1,000 $TCID_{50}$ in 100 μl) were inoculated onto the apical side of the culture and incubated at 37 °C for 1 h. The inoculated viruses were removed and washed twice with Opti-MEM. For collection of the viruses, 100 μl Opti-MEM was applied onto the apical side of the culture and incubated at 37 °C for 10 min. The Opti-MEM was collected and used for RT–qPCR to quantify the viral RNA copy number (see "RT–qPCR" section below). The infection experiments using an airway-on-a-chip system (Fig. 5g) were performed as described above (see "Airway-on-a-chips" section).

## RT–qPCR
RT–qPCR was performed as previously described[2,5,10,24–27,29,38,78]. Briefly, 5 μl culture supernatant was mixed with 5 μl of 2 × RNA lysis buffer [2% Triton X-100 (Nacalai Tesque, Cat# 35501-15), 50 mM KCl, 100 mM Tris-HCl (pH 7.4), 40% glycerol, 0.8 U/μl recombinant RNase inhibitor (Takara, Cat# 2313B)] and incubated at room temperature for 10 min. RNase-free water (90 μl) was added, and the diluted sample (2.5 μl) was used as the template for real-time RT-PCR performed according to the manufacturer's protocol using One Step TB Green PrimeScript PLUS RT-PCR kit (Takara, Cat# RR096A) and the following primers: Forward *N*, 5′-AGC CTC TTC TCG TTC CTC ATC AC-3′; and Reverse *N*, 5′-CCG CCA TTG CCA GCC ATT C-3′. The viral RNA copy number was standardized with a SARS-CoV-2 direct detection RT-qPCR kit (Takara, Cat# RC300A). Fluorescent signals were acquired using a QuantStudio 1 Real-Time PCR system (Thermo Fisher Scientific), QuantStudio 3 Real-Time PCR system (Thermo Fisher Scientific), QuantStudio 5 Real-Time PCR system (Thermo Fisher Scientific), StepOnePlus Real-Time PCR system (Thermo Fisher Scientific), CFX Connect Real-Time PCR Detection system (Bio-Rad), Eco Real-Time PCR System (Illumina), qTOWER3 G Real-Time System (Analytik Jena), Thermal Cycler Dice Real Time System III (Takara) or 7500 Real-Time PCR System (Thermo Fisher Scientific).

## Animal experiments
Animal experiments (Fig. 6 and Supplementary Figs. 5 and 6) were performed as previously described[2,5,10,25–27,38]. Syrian hamsters (Slc:Syrian, male, 4 weeks old) were purchased from Japan SLC Inc. (Shizuoka, Japan). For the virus infection experiments, hamsters were anesthetized by intramuscular injection of a mixture of 0.15 mg/kg medetomidine hydrochloride (Domitor®, Nippon Zenyaku Kogyo), 2.0 mg/kg midazolam (Dormicum®, Fujifilm Wako, Cat# 135-13791) and 2.5 mg/kg butorphanol (Vetorphale®, Meiji Seika Pharma) or 0.15 mg/kg medetomidine hydrochloride, 4.0 mg/kg alphaxaone (Alfaxan®, Jurox) and 2.5 mg/kg butorphanol. Delta, BA.2.75 and XBB.1 (10,000 $TCID_{50}$ in 100 μl) or saline (100 μl) was intranasally inoculated under anesthesia. Oral swabs were collected at the indicated timepoints. Body weight was recorded daily by 7 d.p.i. Enhanced pause (Penh), the ratio of time to peak expiratory follow relative to the total expiratory time (Rpef) were measured every day until 7 d.p.i. (see below). Lung tissues were anatomically collected at 2 and 5 d.p.i. The viral RNA load in the oral swabs and respiratory tissues was determined by RT–qPCR. These tissues were also used for IHC and histopathological analyses (see below).

## Lung function test
Lung function tests (Fig. 6a) were routinely performed as previously described[2,5,10,25–27]. The two respiratory parameters (Penh and Rpef) were measured by using a Buxco Small Animal Whole Body Plethysmography system (DSI) according to the manufacturer's instructions. In brief, a hamster was placed in an unrestrained plethysmography chamber and allowed to acclimatize for 30 s. Then, data were acquired over a 2.5-min period by using FinePointe Station and Review software v2.9.2.12849 (DSI).

## Immunohistochemistry
Immunohistochemical (IHC) analysis (Fig. 6c and Supplementary Fig. 6) was performed as previously described[2,5,10,25–27] using an Autostainer Link 48 (Dako). The deparaffinized sections were exposed to EnVision FLEX target retrieval solution high pH (Agilent, Cat# K8004) for 20 min at 97 °C for activation, and a mouse anti-SARS-CoV-2 N monoclonal antibody (clone 1035111, R&D Systems, Cat# MAB10474-SP, 1:400) was used as a primary antibody. The sections were sensitized using EnVision FLEX for 15 min and visualized by peroxidase-based enzymatic reaction with 3,3'-diaminobenzidine tetrahydrochloride (Dako, Cat# DM827) as substrate for 5 min. The N protein positivity was evaluated by certificated pathologists as previously described[2,5,10,25–27]. Images were incorporated as virtual slides by NDP.scan software v3.2.4 (Hamamatsu Photonics). The N-protein positivity was measured as the area using Fiji software v2.2.0 (ImageJ).

## H&E staining
Haematoxylin and eosin (H&E) staining (Fig. 6d) was performed as previously described[2,5,10,25–27]. Briefly, excised animal tissues were fixed with 10% formalin neutral buffer solution and processed for paraffin embedding. The paraffin blocks were sectioned at a thickness of 3 μm and then mounted on MAS-GP-coated glass slides (Matsunami Glass, Cat# S9901). H&E staining was performed according to a standard protocol.

## Histopathological scoring
Histopathological scoring (Fig. 6e) was performed as previously described[2,5,10,25–27]. The inflammation area in the infected lungs was measured by the presence of the type II pneumocyte hyperplasia. Four hamsters infected with each virus were sacrificed on days 2 and 5 d.p.i., and all four lung lobes, including right upper (anterior/cranial), middle, lower (posterior/caudal), and accessory lobes, were sectioned along with their bronchi. The tissue sections were stained by H&E, and the digital microscopic images were incorporated into virtual slides using NDRscan3.2 software (Hamamatsu Photonics). The color of the images was decomposed by RGB in split channels using Fiji software v2.2.0.

Histopathological scoring was performed as described in the previous studies[2,5,10,25–27]. Pathological features including bronchitis or bronchiolitis, hemorrhage or congestion, alveolar damage with epithelial apoptosis and macrophage infiltration, hyperplasia of type II pneumocytes, and the area of the hyperplasia of large type II pneumocytes were evaluated by certified pathologists and the degree of these pathological findings were arbitrarily scored using four-tiered system as 0 (negative), 1 (weak), 2 (moderate), and 3 (severe). The "large type II pneumocytes" are the hyperplasia of type II pneumocytes exhibiting more than 10-μm-diameter nucleus. Total histology score is the sum of these five indices. In the representative lobe of each lung, the inflammation area with type II pneumocytes was gated by the certificated pathologists on H&E staining, and the indicated area were measured by Fiji software v2.2.0.

## Statistics and reproducibility
Statistical significance was tested using a two-sided Mann–Whitney *U* test, a two-sided Student's *t* test, a two-sided Welch's *t* test, or a two-sided paired *t*-test unless otherwise noted. The tests above were performed using Prism 9 software v9.1.1 (GraphPad Software).

In the time-course experiments (Figs. 3d, 5, 6a–c and 6e, and Supplementary Fig. 2b), a multiple regression analysis including experimental conditions (i.e., the types of infected viruses) as explanatory variables and timepoints as qualitative control variables was performed to evaluate the difference between experimental conditions thorough all timepoints. The initial time point was removed from the analysis. The *P* value was calculated by a two-sided Wald test. Subsequently, familywise error rates (FWERs) were calculated by the Holm method. These analyses were performed in R v4.1.2 (https://www.r-project.org/).

Principal component analysis to representing the antigenicity of the S proteins was performed (Fig. 2g). The $NT_{50}$ values for biological replicates were scaled, and subsequently, principal component analysis was performed using the prcomp function on R v4.1.2 (https://www.r-project.org/).

In Fig. 6c, d and Supplementary Fig. 6, photographs shown are the representative areas of at least two independent experiments by using four hamsters at each timepoint. In Supplementary Fig. 3a, micrographs (scale bars, 50 nm) shown are the representative areas of the XBB.1 S trimer alone or of the XBB.1 S trimer-ACE2 complex in at least two independent datasets using cryo-EM. 2D class images show representative top and side views of the XBB.1 S trimer alone or the XBB.1 S trimer-ACE2 complex from the results of 2D classification with 150 classes using cryoSPARC in each dataset.

## Reporting Summary
Further information on research design is available in the Nature Portfolio Reporting Summary linked to this article.

## Data availability
All databases/datasets used in this study are available from the GISAID database (https://www.gisaid.org) and GenBank database (https://www.gisaid.org; EPI_SET ID: EPI_SET_221223pb [https://epicov.org/epi3/epi_set/221223pb], EPI_SET_221223ew [https://epicov.org/epi3/epi_set/221223ew], EPI_SET_221223yk [https://epicov.org/epi3/epi_set/221223yk], EPI_SET_221222mt [https://epicov.org/epi3/epi_set/221222mt]). Viral genome sequencing data for working viral stocks are available in the Sequence Read Archive (accession ID: PRJDB14899). The atomic coordinates and cryo-EM maps for the structures of the XBB.1 S protein alone closed state 1 (PDB code: 8IOS, EMDB code: 35622), closed state 2 (PDB code: 8IOT, EMDB code: 35623), in complex with human ACE2 one-up state (PDB code: 8IOU, EMDB code: 35624), in complex with human ACE2 two-up state (EMDB code: 35625), and XBB.1 S RBD bound to ACE2 (PDB code: 8IOV, EMDB code: 35626) are available in the Protein Data Bank (www.rcsb.org) and Electron Microscopy Data Bank (www.ebi.ac.uk/emdb/). Source data are provided with this paper.

## Code availability
The computational codes used in the present study and the GISAID supplemental tables are available in the GitHub repository (https://github.com/TheSatoLab/XBB).

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

## Acknowledgements

We would like to thank all members belonging to The Genotype to Phenotype Japan (G2P-Japan) Consortium. We thank Dr. Jin Kuramochi (Interpark Kuramochi Clinic, Japan) for providing patient sera, Dr. Kenzo Tokunaga (National Institute for Infectious Diseases, Japan) and Dr. Jin Gohda (The University of Tokyo, Japan) for providing reagents. We also thank National Institute for Infectious Diseases, Japan for providing clinical isolates of BQ.1.1 (strain TY41-796-P1; GISAID ID: EPI_ISL_15579783) and BA.2 (strain TY40-385; GISAID ID: EPI_ISL_9595859). We appreciate the technical assistance from The Research Support Center, Research Center for Human Disease Modeling, Kyushu University Graduate School of Medical Sciences. We gratefully acknowledge all data contributors, i.e. the Authors and their Originating laboratories responsible for obtaining the specimens, and their Submitting laboratories for generating the genetic sequence and metadata and sharing via the GISAID Initiative, on which this research is based. The super-computing resource was provided by Human Genome Center at The University of Tokyo. This study was supported in part by AMED SCARDA Japan Initiative for World-leading Vaccine Research and Development Centers "UTOPIA" (JP223fa627001, to Kei Sato), AMED SCARDA Program on R&D of new generation vaccine including new modality application (JP223fa727002, to Kei Sato); AMED SCARDA Kyoto University Immunomonitoring Center (KIC) (JP223fa627009, to Takao Hashiguchi); AMED SCARDA World-leading institutes for vaccine research and development Hokkaido Synergy Campus (JP223fa627005 to Takasuke Fukuhara and Keita Matsuno); AMED Research Program on Emerging and Re-emerging Infectious Diseases (JP21fk0108574, to Hesham Nasser; JP21fk0108481, to Akatsuki Saito; JP21fk0108465, to Akatsuki Saito; JP21fk0108493, to Takasuke Fukuhara; JP21fk0108463 to Katsumi Maenaka; JP22fk0108617 to Takasuke Fukuhara; JP22fk0108146, to Kei Sato; JP21fk0108494 to G2P-Japan Consortium, Keita Matsuno, Shinya Tanaka, Terumasa Ikeda, Takasuke Fukuhara, and Kei Sato; JP21fk0108425, to Kazuo Takayama, Akatsuki Saito and Kei Sato; 22fk0108506, to Kazuo Takayama, Akatsuki Saito, and Kei Sato; JP21fk0108432, to Kazuo Takayama, Takasuke Fukuhara and Kei Sato); AMED Research Program on HIV/AIDS (JP22fk0410033, to Akatsuki Saito; JP22fk0410047, to Akatsuki Saito; JP22fk0410055, to Terumasa Ikeda; and JP22fk0410039, to Kei Sato); AMED CRDF Global Grant (JP22jk0210039 to Akatsuki Saito); AMED Japan Program for Infectious Diseases Research and Infrastructure (JP22wm0325009, to Akatsuki Saito; JP22wm0125008 to Keita Matsuno); AMED Japan Initiative for World-leading Vaccine Research and Development Centers (JP223fa627005 to Katsumi Maenaka); AMED Project Focused on Developing Key Technology for Discovering and Manufacturing Drugs for Next-Generation Treatment and Diagnosis (JP20ae0101047 to Katsumi Maenaka); AMED Platform Project for Supporting Drug Discovery and Life Science Research (JP21am0101093 to Katsumi Maenaka); AMED Basis for supporting innovative drug discovery and life science research, phase 2 (JP22ama121037 to Katsumi Maenaka); AMED Advanced Research & Development Programs for Medical Innovation (JP22gm1810004 to Katsumi Maenaka); AMED CREST (JP21gm1610005, to Kazuo Takayama); JST PRESTO (JPMJPR22R1, to Jumpei Ito); JST CREST (JPMJCR20H4, to Kei Sato; JPMJCR20H8, to Takao Hashiguchi); JSPS KAKENHI Grant-in-Aid for Scientific Research C (22K07103, to Terumasa Ikeda); JSPS KAKENHI Grant-in-Aid for Scientific Research B (21H02736, to Takasuke Fukuhara); JSPS KAKENHI Grant-in-Aid for Early-Career Scientists (22K16375, to Hesham Nasser; 20K15767, to Jumpei Ito; 23K14526 to Jumpei Ito); JSPS KAKENHI Grant-in-Aid for Transformative Research Areas B (20H05773 to Takao Hashiguchi); JSPS KAKENHI Grant-in-Aid for Scientific Research on Innovative Areas (JP20H05873, to Katsumi Maenaka); JSPS Core-to-Core Program (A. Advanced Research Networks) (JPJSCCA20190008, to Kei Sato); JSPS Research Fellow DC2 (22J11578, to Keiya Uriu); JSPS Leading Initiative for Excellent Young Researchers (LEADER) (to Terumasa Ikeda);

World-leading Innovative and Smart Education (WISE) Program 1801 from the Ministry of Education, Culture, Sports, Science and Technology (MEXT) (to Naganori Nao); The Cooperative Research Program (Joint Usage/Research Center program) of Institute for Life and Medical Sciences, Kyoto University (to Kei Sato); The Tokyo Biochemical Research Foundation (to Kei Sato); Takeda Science Foundation (to Terumasa Ikeda and Katsumi Maenaka); Mochida Memorial Foundation for Medical and Pharmaceutical Research (to Terumasa Ikeda); The Naito Foundation (to Terumasa Ikeda); Shin-Nihon Foundation of Advanced Medical Research (to Terumasa Ikeda); Waksman Foundation of Japan (to Terumasa Ikeda); an intramural grant from Kumamoto University COVID-19 Research Projects (AMABIE) (to Terumasa Ikeda); Ito Foundation Research Grant R4 (to Akatsuki Saito); International Joint Research Project of the Institute of Medical Science, the University of Tokyo (to Jiri Zahradnik, Daniel Sauter, Terumasa Ikeda, Takasuke Fukuhara, and Akatsuki Saito); the Federal Ministry of Education and Research Germany (BMBF; 01KI20135, to Daniel Sauter); the Canon Foundation Europe (to Daniel Sauter), the Heisenberg Program of the German Research Foundation (DFG; SA 2676/3-1, to Daniel Sauter); grants of the COVID-19 program of the Ministry of Science, Research and the Arts Baden-Württemberg (MWK; K.N.K.C.014 and K.N.K.C.015, to Daniel Sauter); and the project of National Institute of Virology and Bacteriology, Programme EXCELES, funded by the European Union, Next Generation EU (LX22NPO5103, to Jiri Zahradnik).

## Author contributions

J.I. performed bioinformatics, modeling, and statistical analysis.

## Competing interests

Y.Y. and T.N. are founders and shareholders of HiLung, Inc. Y.Y. is a co-inventor of patents (PCT/JP2016/057254; "Method for inducing differentiation of alveolar epithelial cells", PCT/JP2016/059786, "Method of producing airway epithelial cells"). The other authors declare that no competing interests exist.

## Additional information

[1]Department of Microbiology and Immunology, Faculty of Medicine, Hokkaido University, Sapporo, Japan. [2]Institute for Vaccine Research and Development, HU-IVReD, Hokkaido University, Sapporo, Japan. [3]Division of Systems Virology, Department of Microbiology and Immunology, The Institute of Medical Science, The University of Tokyo, Tokyo, Japan. [4]Graduate School of Medicine, The University of Tokyo, Tokyo, Japan. [5]Department of Biomolecular Sciences, Weizmann Institute of Science, Rehovot, Israel. [6]First Medical Faculty at Biocev, Charles University, Vestec-Prague, Czechia. [7]Division of Risk Analysis and Management, International Institute for Zoonosis Control, Hokkaido University, Sapporo, Japan. [8]Laboratory of Biomolecular Science and Center for Research and Education on Drug Discovery, Faculty of Pharmaceutical Sciences, Hokkaido University, Sapporo, Japan. [9]Division of Molecular Virology and Genetics, Joint Research Center for Human Retrovirus infection, Kumamoto University, Kumamoto, Japan. [10]Department of Clinical Pathology, Faculty of Medicine, Suez Canal University, Ismailia, Egypt. [11]Department of Veterinary Science, Faculty of Agriculture, University of Miyazaki, Miyazaki, Japan. [12]Graduate School of Medicine and Veterinary Medicine, University of Miyazaki, Miyazaki, Japan. [13]Department of Cancer Pathology, Faculty of Medicine, Hokkaido University, Sapporo, Japan. [14]Medical Research Council-University of Glasgow Centre for Virus Research, Glasgow, UK. [15]Division of International Research Promotion, International Institute for Zoonosis Control, Hokkaido University, Sapporo, Japan. [16]One Health Research Center, Hokkaido University, Sapporo, Japan. [17]Division of Molecular Pathobiology, International Institute for Zoonosis Control, Hokkaido University, Sapporo, Japan. [18]Center for iPS Cell Research and Application (CiRA), Kyoto University, Kyoto, Japan. [19]Institute for Chemical Reaction Design and Discovery (WPI-ICReDD), Hokkaido University, Sapporo, Japan. [20]Laboratory of Medical Virology, Institute for Life and Medical Sciences, Kyoto University, Kyoto, Japan. [21]Department of Medicinal Sciences, Graduate School of Pharmaceutical Sciences, Kyushu University, Fukuoka, Japan. [22]Graduate School of Frontier Sciences, The University of Tokyo, Kashiwa, Japan. [23]Institute for Medical Virology and Epidemiology of Viral Diseases, University Hospital Tübingen, Tübingen, Germany. [24]Institute for Genetic Medicine, Hokkaido University, Sapporo, Japan. [25]Tokyo Metropolitan Institute of Public Health, Tokyo, Japan. [26]HiLung, Inc., Kyoto, Japan. [27]Global Station for Biosurfaces and Drug Discovery, Hokkaido University, Sapporo, Japan. [28]Division of Pathogen Structure, International Institute for Zoonosis Control, Hokkaido University, Sapporo, Japan. [29]CREST, Japan Science and Technology Agency, Kawaguchi, Japan. [30]AMED-CREST, Japan Agency for Medical Research and Development (AMED), Tokyo, Japan. [31]Laboratory of Virus Control, Research Institute for Microbial Diseases, Osaka University, Suita, Japan. [32]Center for Animal Disease Control, University of Miyazaki, Miyazaki, Japan. [33]International Collaboration Unit, International Institute for Zoonosis Control, Hokkaido University, Sapporo, Japan. [34]International Research Center for Infectious Diseases, The Institute of Medical Science, The University of Tokyo, Tokyo, Japan. [35]International Vaccine Design Center, The Institute of Medical Science, The University of Tokyo, Tokyo, Japan. [36]Collaboration Unit for Infection, Joint Research Center for Human Retrovirus infection, Kumamoto University, Kumamoto, Japan. [41]These authors contributed equally: Tomokazu Tamura, Jumpei Ito, Keiya Uriu, Jiri Zahradnik, Izumi Kida, Yuki Anraku, Hesham Nasser, Maya Shofa, Yoshitaka Oda, Spyros Lytras. ✉e-mail: tanaka@med.hokudai.ac.jp; matsuk@czc.hokudai.ac.jp; kazuo.takayama@cira.kyoto-u.ac.jp; KeiSato@g.ecc.u-tokyo.ac.jp

## The Genotype to Phenotype Japan (G2P-Japan) Consortium

Hayato Ito[1], Naoko Misawa[3], Izumi Kimura[3], Mai Suganami[3], Mika Chiba[3], Ryo Yoshimura[3], Kyoko Yasuda[3], Keiko Iida[3], Naomi Ohsumi[3], Adam P. Strange[3], Otowa Takahashi[9], Kimiko Ichihara[9], Yuki Shibatani[11], Tomoko Nishiuchi[11], Marie Kato[13], Zannatul Ferdous[13], Hiromi Mouri[13], Kenji Shishido[13], Hirofumi Sawa[2,15,16,17], Rina Hashimoto[18], Yukio Watanabe[18], Ayaka Sakamoto[18], Naoko Yasuhara[18], Tateki Suzuki[20], Kanako Kimura[20], Yukari Nakajima[20], So Nakagawa[37], Jiaqi Wu[37], Kotaro Shirakawa[38], Akifumi Takaori-Kondo[38], Kayoko Nagata[38], Yasuhiro Kazuma[38], Ryosuke Nomura[38], Yoshihito Horisawa[38], Yusuke Tashiro[38], Yugo Kawai[38], Takashi Irie[39], Ryoko Kawabata[39], Chihiro Motozono[40], Mako Toyoda[40] & Takamasa Ueno[40]

[37]Tokai University School of Medicine, Isehara, Japan. [38]Kyoto University, Kyoto, Japan. [39]Hiroshima University, Hiroshima, Japan. [40]Kumamoto University, Kumamoto, Japan.

