## [Peer Review File · Nature Communications]

Reviewer comments, first round -

Reviewer #1 (Remarks to the Author):

The study by Tamara and coworkers describes viral characteristics of XBB, a descendent of BA.2, that is quickly expanding in multiple regions across the world, together with BQ.1 lineages. While BQ.1, the currently dominant strain acquired all convergent amino acid substitutions by diversification, XBB acquired convergent mutations by recombination. This study is an extensive and well performed characterization of XBB. The authors addressed viral phylogenetic characteristics, evasion from neutralizing antibodies from SARS-CoV-2 infected-patients, viral binding affinity, fusogenicity, infectivity in multiple in vitro cell culture systems as well as viral spread and intrinsic pathogenicity in a hamster model. They demonstrated that XBB is antigenically different from other Omicron subvariants and evades BA.2/5 infection-induced herd immunity due to the presence of mutations in the RBD and NTD region. They showed that XBB harbors immune escape-associated substitutions, including Y144del and F486S, as well as, infectivity-enhancing substitutions, such as V83A. Several aspects of the manuscript are of potential interest to public health besides previous reports in the literature in the same topic. However, besides the potential interest, the manuscript raises the following concerns:

- 1- Fig. 1: Authors compared XBB to BQ.1 characteristics throughout the manuscript, because of that, it would be helpful to incorporate BQ1 mutations on Figure 1b panel for reference.
 - 2- Fig. 2: Authors used a HIV-1 based pseudovirus to address sera immune resistance. While XBB exhibits a marked resistance to breakthrough infection sera from infected patients and sera obtained from infected hamsters, the immune resistance conferred by single substitutions is minor. As indicated by the authors, this suggests cooperative contribution of mutations (that can be synergetic, antagonistic, or neutral) on antibody-resistance. As 1) all tested individual substitutions have minor impact in sera resistance, 2) multiple individual mutations are shared between BA.2.75 and XBB and have a significant in sera resistance, 3) BA.2.75 did not exhibit significant resistance when compared to BA.2, 4) Substitutions V83 and 252 and 144del deletions are the only mutations observed in XBB, not shared with BA.2.75 with significant impact in sera resistance: authors could extend these analyses and individually introduce these substitutions or deletion in the BA.2.75 backbone to investigate/confirm cooperative contribution and the substantial reduction observed with XBB. Also, for clarity, authors should include a brief introduction of the hamster model before starting to describe antigenicity of XBB S using sera obtained from infected hamsters. Authors described XBB as the most profoundly resistant variant to BA.2/5 breakthrough infection, however, while BQ.1. resistance has been evaluated using sera from infected hamsters, the neutralization assays were not performed with the human samples.
 - 3- Fig.3: Infectivity of XBB was addressed using pseudovirus and HOS-ACE2-TMPRSS2 cells. The results indicate that the XBB Spike augments its infectious potential through substitutions in the RBD and NTD. Authors show no impact of TMPRSS2 expression in pseudovirus infection, however in vitro culture with WT viruses suggests differently. Could the authors indicate if pseudovirus-based assays have demonstrated a good correlation with the WT virus-based? On panel 3e, the S-based fusion assay showed that the XBB S is significantly more fusogenic than BA.2 S and BA.2.75 S. Since, BA.2.75 and XBB share multiple mutations, the author could include BA.2.75 results in the individual mutations panels for reference and better understanding of the mutation's impact. Finally, the authors measured RNA load from infected cultures to complement fusogenicity assays. Could the authors extend this analysis and measure infectious viral particles in the supernatant to complement infectivity results?
- Minor: Extended Data Fig.S1a is not cited in the main text.

Reviewer #2 (Remarks to the Author):

This multidisciplinary, collaborative manuscript provides an overview of the XBB variant from a variety of perspectives, including phylogenetic, epidemiologic, and virological assessments.

Authors utilize these diverse experimental approaches to speculate on the geographical origins of the variant, the recombination events that led to the emergence of the variant, and the relative level of pathogenicity using different in vitro and in vivo models. This consortium of collaborators has published similar broad-ranging assessments of different emerging variants employing a similar set of experimental tools. The topic of study is of high interest to the field and would be applicable to a wide range of scientists due to the diversity of approaches examined. That said, there are areas of the manuscript which could benefit from improvement, clarity, and rigor.

Major comments:

1. The discussion section insufficiently references literature outside of what has been published by the collaborators themselves. The majority of references cite to previous work from the participating groups involved in the collaboration but only provide (by my estimation) 3-4 references to outside research groups. While there may be a paucity of XBB variant publications available to cite given the recent nature of this variant emergence, results presented should still be contextualized much more rigorously with the expansive body of literature on emerging Omicron variants, both with the number of references to support statements, and citing work outside of the specific methodology employed in these studies.
2. It is currently unclear why authors switched from BA.2 to Delta variant for virological characterizations. Please better justify in the text why this experimental decision was performed.
3. All virological characterizations in this study appear to report viral RNA levels, and not infectious virus titers. Why were viral titers not quantified alongside RNA levels in any assay? Is the ratio of RNA:TCID50 the same for each variant examined? Without this information it is challenging for a reader to understand how viral quantification levels translate to infectious virus loads in different cell types in vitro and in vivo.
4. For airway-on-chips results presented in figure 3l, what is causing the increased permeability of this barrier resulting in greater detection of selected viruses in the basolateral compartment? Is this due to breakdown of tight junctions between cells, increased apoptosis/necrosis following viral infection, inflammatory host responses, etc? It's a bit confusing that the authors transition from this finding to the conclusion that these results contribute to XBB having higher fusogenicity relative to other variants without providing a mechanistic explanation (lines 324-332).

Other comments:

1. Abstract, line 87, "...the most profoundly resistant variant...ever" is a bit bombastic, suggest rephrasing to "...the most profoundly resistant variant...to date".
2. For statistical analyses of data in Figure 3, please report all pairwise statistical results for panels f-k, not just the ones against XBB (or otherwise provide the reader with an understanding of the statistical results against all variables tested).
3. Figure 3l, please label Delta variant in the legend.
4. Figure 4c, please provide an uninfected staining control.

Reviewer #3 (Remarks to the Author):

Major points:

The tree in figure 1a does not look like other trees that I have seen of the Omicron sublineages. Importantly, BA.5 (and BA.4) is NOT a descendent of BA.2, it is a sister lineage, hence why it is not BA.2.XX. Comparing this tree to, for example, the one in Tegally et al 2022 (<https://www.nature.com/articles/s41591-022-01911-2>) shows that the branching pattern in this tree is unexpected.

Given the small size of the alignment, and the importance of an accurate topology, I don't understand why the authors have decided to use maximum likelihood approaches compared to a Bayesian analysis. Given that they then perform a discrete trait analysis, I would recommend the authors re-analyse the data using BEAST and perform the DTA jointly with the topology and timing estimation as compared to each step independently in an ML framework. This would also provide a more reliable root to the tree rather than simply outgroup rooting.

Finally, I don't think the phylogeographic inference is reliable with this dataset. It is a very small dataset, and I would prefer some more principled method of downsampling taking location into account rather than simply 100 or 20 random samples of each lineage. Further, why were India, Bangladesh and Singapore in particular used as individual states? It may well be that XBB originated in India, but there is no real acknowledgement or attempt at mitigation (that I can see from these methods) of the effect of sampling bias on discrete trait reconstruction.

My suggestions are therefore as follows: the authors regenerate the dataset using more sequences, and downsampled based on some metric that takes time and location into account (eg 10 samples every week from each continent or something – they should investigate the literature to see what other downsampled studies use). They should then use BEAST to infer the topology, timing, and location of the nodes at the same time (possible with less than ~3000 sequences or so depending on their computational resources – if the dataset is larger, infer the time tree first and then perform the DTA on a set of empirical trees). Downsampling is a difficult issue, and they may want to consider simply removing the DTA portion of the manuscript, given that it is mostly a laboratory-based study.

Minor points

Line 94 – the other variants have not increased fitness through single mutations, they are groups of mutations. It's a good point to make, just needs a bit of rephrasing. Same in line 181, and line 395.

Line 100 – data becomes out of date very rapidly, especially at the leading edge of the pandemic. I would rephrase this section about BQ.1.1 as the increase in frequency of it has plateaued at about 25% worldwide, and perhaps phrase it more historically so that it is not rapidly out of date.

Line 121 – you can also now add that XBB.1.1 has become an issue in the northeast of the US

Line 133 – it is better to say that BJ.1 and BA.2.75 were first detected in South Asia, as there doesn't appear to have been an analysis to confirm they originated there specifically.

Line 400 – it's worth pointing out that geographic distance doesn't matter as much as distance through flight networks. Eg the UK is not geographically close to India, but has very strong travel connections and so Delta variant was sampled early in the UK compared to lots of the rest of the world. Further, an immune escape variant could have variation in viral fitness in different regions due to variations in infection waves, vaccination etc. This point therefore needs more nuance.

Reviewer #1 (Remarks to the Author):

The study by Tamara and coworkers describes viral characteristics of XBB, a descendent of BA.2, that is quickly expanding in multiple regions across the world, together with BQ.1 lineages. While BQ.1, the currently dominant strain acquired all convergent amino acid substitutions by diversification, XBB acquired convergent mutations by recombination. This study is an extensive and well performed characterization of XBB. The authors addressed viral phylogenetic characteristics, evasion from neutralizing antibodies from SARS-CoV-2 infected-patients, viral binding affinity, fusogenicity, infectivity in multiple in vitro cell culture systems as well as viral spread and intrinsic pathogenicity in a hamster model. They demonstrated that XBB is antigenically different from other Omicron subvariants and evades BA.2/5 infection-induced herd immunity due to the presence of mutations in the RBD and NTD region. They showed that XBB harbors immune escape-associated substitutions, including Y144del and F486S, as well as, infectivity-enhancing substitutions, such as V83A. Several aspects of the manuscript are of potential interest to public health besides previous reports in the literature in the same topic.

However, besides the potential interest, the manuscript raises the following concerns:

Our reply:

First of all, we would like to thank the reviewer for the positive comments. To complete this study, we have addressed reviewer concerns. Our point-by-point replies are summarized below.

1- Fig. 1: Authors compared XBB to BQ.1 characteristics throughout the manuscript, because of that, it would be helpful to incorporate BQ1 mutations on Figure 1b panel for reference.

Our reply:

In accordance with this suggestion, we have included BQ.1.1 in the heatmap shown in **Fig. 1b**.

2- Fig. 2: Authors used a HIV-1 based pseudovirus to address sera immune resistance. While XBB exhibits a marked resistance to breakthrough infection sera from infected patients and sera obtained from infected hamsters, the immune resistance conferred by single substitutions is minor. As indicated by the authors, this suggests cooperative contribution of

mutations (that can be synergetic, antagonistic, or neutral) on antibody-resistance. As 1) all tested individual substitutions have minor impact in sera resistance, 2) multiple individual mutations are shared between BA.2.75 and XBB and have a significant impact in sera resistance, 3) BA.2.75 did not exhibit significant resistance when compared to BA.2, 4) Substitutions V83 and 252 and 144del deletions are the only mutations observed in XBB, not shared with BA.2.75 with significant impact in sera resistance: authors could extend these analyses and individually introduce these substitutions or deletion in the BA.2.75 backbone to investigate/confirm cooperative contribution and the substantial reduction observed with XBB.

Our reply:

A recent study by Cao et al. showed the importance of V83A (Cao et al., Nature, 2023; PMID 36535326). The figure is shown below:

<https://www.nature.com/articles/s41586-022-05644-7>

Additionally, from the results of hamster sera (**Fig. 2g** of the revised manuscript), it is clear that the antigenicity between BA.2.75 and XBB.1 is different. It is interesting to determine which mutations define the immune resistance of XBB.1, but from the results of BA.2-based neutralization assays, it can be inferred that it is not determined by a single amino acid substitution. Moreover, our additional data using vaccine sera (**Fig. 2c-e** of the revised manuscript) further show the complicatedness of how XBB.1 exhibits immune resistance.

Also, for clarity, authors should include a brief introduction of the hamster model before starting to describe antigenicity of XBB S using sera obtained from infected hamsters. Authors described XBB as the most profoundly resistant variant to BA.2/5 breakthrough infection, however, while BQ.1. resistance has been evaluated using sera from infected hamsters, the neutralization assays were not performed with the human samples.

Our reply:

We added the description for rationale to use sera of infected laboratory animals, whose immune history can be fully controlled, for the antigenicity test in the revised manuscript (lines 264-270, page 8). Sera collected from humans infected with only a sole VOC are becoming less available and may be nearly impossible to obtain in the future. Thus, our study using hamster sera demonstrates the usefulness of laboratory animal sera for the antigenicity assays as performed in influenza research.

3- Fig.3: Infectivity of XBB was addressed using pseudovirus and HOS-ACE2-TMPRSS2 cells. The results indicate that the XBB Spike augments its infectious potential through substitutions in the RBD and NTD. Authors show no impact of TMPRSS2 expression in pseudovirus infection, however in vitro culture with WT viruses suggests differently. Could the authors indicate if pseudovirus-based assays have demonstrated a good correlation with the WT virus-based?

Our reply:

Thanks for this important comment. The data shown in **Fig. 3c** of the previous manuscript was not consistent with the other data, and has therefore been removed from the revised manuscript to avoid confusion.

On panel 3e, the S-based fusion assay showed that the XBB S is significantly more fusogenic than BA.2 S and BA.2.75 S. Since, BA.2.75 and XBB share multiple mutations, the author could include BA.2.75 results in the individual mutations panels for reference and better understanding of the mutation's impact.

Our reply:

In **Fig. 3d** (**Fig. 3e** of the original manuscript), greater fusogenicity of XBB than BA.2.75 is

shown. Additionally, the data of BA.2-based point mutants that harbor the mutations shared with XBB and BA.2.75 are shown in **Extended Data Fig. 2**, and the BA.2-based point mutants that harbor XBB-specific mutations are shown in **Fig. 3d**. We think there is no necessity to combine these, and the current data are sufficient to interpret and explain the results.

Finally, the authors measured RNA load from infected cultures to complement fusogenicity assays. Could the authors extend this analysis and measure infectious viral particles in the supernatant to complement infectivity results?

Our reply:

This concern was also raised by another reviewer. To satisfy this concern, infectious viral titers were measured with following samples; top and bottom channels of airway-on-a-chip at 6 d.p.i. (**Extended Fig. 5a**) and lung periphery of infected hamsters (**Extended Fig. 5b**) to demonstrate that the viral RNA load we measured reflect viral titer. These extended results clearly demonstrate that the viral particles in the cell culture supernatant or tissue could complement infectivity results.

Minor: Extended Data Fig.S1a is not cited in the main text.

Our reply:

The missing citation has been added.

Reviewer #2 (Remarks to the Author):

This multidisciplinary, collaborative manuscript provides an overview of the XBB variant from a variety of perspectives, including phylogenetic, epidemiologic, and virological assessments. Authors utilize these diverse experimental approaches to speculate on the geographical origins of the variant, the recombination events that led to the emergence of the variant, and the relative level of pathogenicity using different in vitro and in vivo models. This consortium of collaborators has published similar broad-ranging assessments of different emerging variants employing a similar set of experimental tools. The topic of study is of high interest to the field and would be applicable to a wide range of scientists due to the diversity of approaches examined. That said, there are areas of the manuscript which could benefit from improvement, clarity, and rigor.

Our reply:

We would like to thank the reviewer for the positive and critical comments. To complete this study, we have addressed reviewer concerns. Our point-by-point replies are summarized below.

Major comments:

1. The discussion section insufficiently references literature outside of what has been published by the collaborators themselves. The majority of references cite to previous work from the participating groups involved in the collaboration but only provide (by my estimation) 3-4 references to outside research groups. While there may be a paucity of XBB variant publications available to cite given the recent nature of this variant emergence, results presented should still be contextualized much more rigorously with the expansive body of literature on emerging Omicron variants, both with the number of references to support statements, and citing work outside of the specific methodology employed in these studies.

Our reply:

Thank you very much for providing this important suggestion. Accordingly, we have referred the papers reported from other groups.

2. It is currently unclear why authors switched from BA.2 to Delta variant for virological

characterizations. Please better justify in the text why this experimental decision was performed.

Our reply:

Thanks for following our G2P-Japan studies! Since the Delta variant is the most pathogenic variant to date, analyzing whether the pathogenicity of the variant of concern (XBB.1 in this case) is greater than that of the Delta variant is an important point. BA.2 is less pathogenic, thus if we compare the pathogenicity of the variant of concern to that of BA.2, what we can say from this comparison is not so significant. We have explained it in more detail in the revised manuscript (lines 437-438, page 13).

3. All virological characterizations in this study appear to report viral RNA levels, and not infectious virus titers. Why were viral titers not quantified alongside RNA levels in any assay? Is the ratio of RNA:TCID50 the same for each variant examined? Without this information it is challenging for a reader to understand how viral quantification levels translate to infectious virus loads in different cell types in vitro and in vivo.

Our reply:

Because recent SARS-CoV-2 studies including ours reported RNA levels as indicators of infectious viral titers without any major inconsistencies, we originally employed the quantity of RNAs in our assays. To prove the ratio of RNA:TCID50 is not different for each variant including the latest XBB variant and the significant differences shown in the present study can be supported by viral titers too, we examined following samples: top and bottom channels of airway-on-a-chip at 6 d.p.i. (**Extended Data Fig. 5a**, RNA load was shown in **Fig. 5h** of revised manuscript) and also lung periphery of infected hamsters (**Extended Data Fig. 5b**, RNA load was shown in the right panel, **Fig. 6b**). Our results clearly demonstrate that the viral particles in the cell culture supernatant or tissue could complement infectivity results.

4. For airway-on-chips results presented in figure 3l, what is causing the increased permeability of this barrier resulting in greater detection of selected viruses in the basolateral compartment? Is this due to breakdown of tight junctions between cells, increased apoptosis/necrosis following viral infection, inflammatory host responses, etc? It's a bit confusing that the authors transition from this finding to the conclusion that these results contribute to XBB having higher fusogenicity relative to other variants without providing a

mechanistic explanation (lines 324-332).

Our reply:

In our previous report (PMID: 36129989), we analyzed the cause of the increased permeability of this barrier. VE-cadherin staining images show that endothelial adherent junctions are weakened by SARS-CoV-2 infection (A). On the other hand, the cell viability of endothelial cells is not decreased (B). Gene expression levels of IL-6, VCAM-1, and ICAM-1 were increased in endothelial cells of airway-on-a-chips, suggesting the induction of inflammatory host responses was caused by the viral infection (C). From the above, it is suggested that XBB has a high barrier-breaking ability by weakening endothelial adherens junctions.

Modified from our previous report (PMID: 36129989)

Other comments:

1. Abstract, line 87, "...the most profoundly resistant variant...ever" is a bit bombastic, suggest rephrasing to "...the most profoundly resistant variant...to date".

Our reply: Edited as suggested.

2. For statistical analyses of data in Figure 3, please report all pairwise statistical results for panels f-k, not just the ones against XBB (or otherwise provide the reader with an understanding of the statistical results against all variables tested).

Our reply: We added a statistical analysis comparing BA.2 and XBB in **Fig. 3d–j** in the revised manuscript.

3. Figure 3l, please label Delta variant in the legend.
4. Figure 4c, please provide an uninfected staining control.

Our reply: We have applied both suggestions.

Reviewer #3 (Remarks to the Author):

Major points:

The tree in figure 1a does not look like other trees that I have seen of the Omicron sublineages. Importantly, BA.5 (and BA.4) is NOT a descendent of BA.2, it is a sister lineage, hence why it is not BA.2.XX. Comparing this tree to, for example, the one in Tegally et al 2022 (<https://www.nature.com/articles/s41591-022-01911-2>) shows that the branching pattern in this tree is unexpected.

Our reply:

Thank you for the comments on this critical point. Although the BEAST tree in Tegally et al. supports that BA.4 and BA.5 are sister clades of BA.2, it remains controversial whether BA.4 and BA.5 are sister clades or descendants of BA.2. For example, Nextstrain explains that BA.4 and BA.5 are descendants of BA.2 (<https://covariants.org/>). Furthermore, phylogenetic trees in several previous works including ours support the tree topology that BA.4 and BA.5 are descendants of BA.2 (PMID: 36198317; PMID: 35856385; PMID: 35790190; PMID: 36270286). Indeed, in the limitation section of the Tegally et al. paper, the authors declare that their ML tree suggests that BA.4 and BA.5 are descendants of BA.2 as follows: “*Although the Bayesian phylogenetic methods employed here suggest that BA.4 and BA.5 are independent lineages that originated around the same time as BA.1–BA.3, maximum likelihood estimations suggest that they could have descended from BA.2.*”

As mentioned in the Tegally et al. paper, recombination events may have contributed to the Omicron evolution. Therefore, it might be incorrect to represent the phylogeny of Omicron lineages as a single tree. However, we assume that it is beneficial to summarize diverse Omicron subvariants appearing in the present study as a phylogenetic tree for readers who are not familiar with these subvariants. Therefore, we retained the phylogenetic tree shown in **Fig. 1a** in the revised manuscript. Instead, we removed the sentences claiming that BA.4 and BA.5 are descendants of BA.2 from the revised manuscript.

Fig. R1. The phylogenetic relationship of variants summarized in Nextstrain (<https://covariants.org/>).

Given the small size of the alignment, and the importance of an accurate topology, I don't understand why the authors have decided to use maximum likelihood approaches compared to a Bayesian analysis. Given that they then perform a discrete trait analysis, I would recommend the authors re-analyse the data using BEAST and perform the DTA jointly with the topology and timing estimation as compared to each step independently in an ML framework. This would also provide a more reliable root to the tree rather than simply outgroup rooting.

Our reply:

We agree that a BEAST phylogenetic inference should provide a more accurate topology and better time estimates. We have now replaced the ML time-calibrated inference with a BEAST MCC phylogeny. As mentioned below, we have decided to remove the location discrete trait analysis from the manuscript due to the likely sampling bias affecting the inference, so this was not included in the BEAST analysis. The overall tree topology and dating inferences were, reassuringly, quite consistent between our previous ML approach and BEAST. We now date the XBB tMRCA to July 7, 2022 (instead of August 5, 2022) and the XBB-BJ.1 split to June 11, 2022 (instead of July 22, 2022). With the BEAST approach, the window of time for the recombination event has been shifted by about a month to the past compared to the ML estimates. The Bayesian approach also provides us with confidence intervals around the inferred dates that we report in the updated manuscript:

“Using the longer 5' end non-recombinant part of these genomes, we estimated the emergence date of XBB using a Bayesian tip-dated phylogenetic inference (see Methods) (Fig. 1d). Our analysis suggests that the XBB clade's most recent common ancestor (tMRCA) existed at the start of July 2022 (median posterior date: July 7, 2022; 95% HPD confidence intervals: from June 10, 2022, to July 29, 2022). We also date the tMRCA between the XBB and BJ.1 lineages at the start of June 2022 (median posterior date: June 11, 2022; 95% HPD intervals: from May 22, 2022, to June 26, 2022) (Fig. 1d). Together, our analyses suggest that XBB emerged through the recombination of two co-circulating lineages, BJ.1 and BM.1.1.1, during the summer of 2022.”

Even though the BEAST inference does not change the general timing of the recombination event (summer of 2022), this approach hopefully provides more reliable dating estimates for our analysis.

Finally, I don't think the phylogeographic inference is reliable with this dataset. It is a very

small dataset, and I would prefer some more principled method of downsampling taking location into account rather than simply 100 or 20 random samples of each lineage. Further, why were India, Bangladesh and Singapore in particular used as individual states? It may well be that XBB originated in India, but there is no real acknowledgement or attempt at mitigation (that I can see from these methods) of the effect of sampling bias on discrete trait reconstruction.

My suggestions are therefore as follows: the authors regenerate the dataset using more sequences, and downsampled based on some metric that takes time and location into account (eg 10 samples every week from each continent or something – they should investigate the literature to see what other downsampled studies use). They should then use BEAST to infer the topology, timing, and location of the nodes at the same time (possible with less than ~3000 sequences or so depending on their computational resources – if the dataset is larger, infer the time tree first and then perform the DTA on a set of empirical trees). Downsampling is a difficult issue, and they may want to consider simply removing the DTA portion of the manuscript, given that it is mostly a laboratory-based study.

Our reply:

Thank you for sharing your concerns about the phylogeographic analysis to infer the emergence place of XBB. We agree that we cannot eliminate the possibility that unbiased sampling affects the inferred result and that the phylogeographic analysis is not the main focus of the present study. Therefore, we have decided to remove the result of the phylogeographic analysis from the revised manuscript.

Minor points

Line 94 – the other variants have not increased fitness through single mutations, they are groups of mutations. It's a good point to make, just needs a bit of rephrasing. Same in line 181, and line 395.

Our reply: We agree and have rephrased these points (line 113, page 4; line 202, page 7; line 500, page 15).

Line 100 – data becomes out of date very rapidly, especially at the leading edge of the pandemic. I would rephrase this section about BQ.1.1 as the increase in frequency of it has

plateaued at about 25% worldwide, and perhaps phrase it more historically so that it is not rapidly out of date.

Our reply: Thank you for the comment. We have specified which time points we are referring to throughout the manuscript.

Line 121 – you can also now add that XBB.1.1 has become an issue in the northeast of the US

Our reply: In accordance with this suggestion, we have added the description of the epidemics of XBB.1.5 in the **Discussion** section (lines 509–511, page 15).

Line 133 – it is better to say that BJ.1 and BA.2.75 were first detected in South Asia, as there doesn't appear to have been an analysis to confirm they originated there specifically.

Our reply: We have rephrased the point mentioned (lines 153–156, pages 6).

Line 400 – it's worth pointing out that geographic distance doesn't matter as much as distance through flight networks. Eg the UK is not geographically close to India, but has very strong travel connections and so Delta variant was sampled early in the UK compared to lots of the rest of the world. Further, an immune escape variant could have variation in viral fitness in different regions due to variations in infection waves, vaccination etc. This point therefore needs more nuance.

Our reply: We agree with this comment and have revised our claim relating to the point above in the **Discussion** section (lines 505–507, page 15).

Reviewer comments, second round -

Reviewer #1 (Remarks to the Author):

All concerns previously raised have been addressed satisfactorily and I do not have any further comments.

Reviewer #2 (Remarks to the Author):

Authors have addressed all comments raised during peer review to my satisfaction; no new comments.

Reviewer #3 (Remarks to the Author):

Thanks for explaining the BA.2 to BA.5 relationship in other papers. While the difference between a BEAST topology and an ML topology is partly why I would personally use the former, this is an acceptable response to me (and I understand why they don't want to get into it in this paper as it's not the point of the research). The authors should say explicitly that it is a maximum likelihood phylogeny in the figure legend for 1A to make this point clear.

Thank you to the authors for doing the BEAST analysis – I agree that it is reassuring that it does not change the main conclusions, but I think this is a better estimate (and as they say, they now have confidence intervals.)

I'm happy with their responses to all the other comments!

-

Author response

[redacted]

Together with the three reviewers of this paper, we have strived to ensure the validity of the information and statements in this paper. All reviewers have accepted our modifications made based on their feedback and have no further comments. In particular with reviewer #3, we worked together to thoroughly confirm that despite having different preferences in approach, our phylogenetic results are consistent across multiple methodologies.

[redacted]